



**O₃-precursor relationship over multiple patterns of time scale:**
**A case study in Zibo, Shandong Province, China**
Zhensen Zheng[1], Kangwei Li[2*], Bo Xu[3], Jianping Dou[4], Liming Li[1], Guotao Zhang[1],
Shijie Li[1], Chunmei Geng[1], Wen Yang[1], Merched Azzi[5], Zhipeng Bai[1*]
[1] State Key Laboratory of Environmental Criteria and Risk Assessment, Chinese
Research Academy of Environmental Sciences, Beijing 100012, China
[2] Univ Lyon, Université Claude Bernard Lyon 1, CNRS, IRCELYON, F-69626
Villeurbanne, France
[3] Zibo Eco-Environmental Monitoring Center, Zibo 255000, China
[4] Zibo Ecological Environment Quality Control Service Center, Zibo 255095,
China
[5] New South Wales Department of Planning, Industry and Environment, PO Box 29,
Lidcombe, NSW 1825, Australia
**\*Corresponding authors**:
Kangwei Li (likangweizju@foxmail.com); Zhipeng Bai (baizp@craes.org.cn)
**Abstract.** In this study, we developed an approach that integrating multiple patterns
of time scale for box modeling (MCMv3.3.1) to better understand the O₃-precursor
relationship through multiple-site and continuous observations. A five-month field
campaign was conducted in the summer of 2019 to investigate the ozone formation
chemistry at three sites in a major prefecture-level city (Zibo) in Shandong province of
northern China. It was found that the relative incremental reactivity (RIR) of major
precursor groups (e.g., anthropogenic volatile organic compound (AVOC), NOₓ) were
overall consistent along with time scale (four patterns: five-month, monthly, weekly,
and daily) varied from wider to narrower, though the magnitude of RIR varied at each
site. The time series of the photochemical regime (using RIR$_{NOx}$/RIR$_{AVOC}$ as indicator)
in weekly or daily patterns further showed varied magnitude but a synchronous
temporal trend among the three sites. The derived RIR ranking (top 10) of individual
AVOC species showed consistency at three averaged patterns (i.e., five-month, monthly,
and weekly). It was further found that the campaign-averaging photochemical regimes
showed overall consistency but non-negligible variability among the four patterns of
time scale, which was mainly due to the embedded uncertainty in model input dataset
when averaging individual daily pattern into different timescales. This implies that
integrating multiple patterns of time scale is useful to derive reliable and robust O₃-
precursor relationship. Our results highlight the importance of quantifying the impact



of different time scales to constrain the photochemical regime, which can formulate
more accurate policy-relevant guidance for $O_3$ pollution control.

## 41  1 Introduction

Since 2013, the ambient $PM_{2.5}$ concentration in China has dramatically declined
by implementing Clean Air Action (Lu et al., 2018; Wang et al., 2020b; Zhang et al.,
2019). However, national ground surface ozone concentrations increased over the same
period (Xue et al., 2020) and became a major air quality problem that needed to be
addressed in China (Li et al., 2019; Wang et al., 2019). It is well-known that ground
surface ozone is formed mainly by complex nonlinear photochemical oxidation of
volatile organic compounds (VOCs) in the presence of nitrogen oxides ($NO_x$ = NO +
$NO_2$) and sunlight (Blanchard, 2000; Hidy, 2000; Kleinman, 2000), which adversely
influences human health, vegetation and corps (Brunekreef and Holgate, 2002;
Vingarzan, 2004).
Given the non-linearity of ozone pollution and complex process involved in it,
challenges in mitigating its severity lies primarily in comprehensively understanding of
$O_3$-precursor relationship (Su et al., 2018a; Tan et al., 2018a). It is commonly
recognized that regional-scale air quality models and the 0-D box model are two
mainstream approaches to investigate the increasingly severe ozone problem
(Blanchard, 2000; Cardelino and Chameides, 1995; Hidy, 2000; Liu et al., 2019). The
0-D box model is an advanced observation-based model that implemented with gas-
phase chemical mechanism, and has been widely used to diagnose $O_3$-precursor
relationship in various locations (Liu et al., 2021a; Sun et al., 2016; Tan et al., 2019b;
Xue et al., 2014a; Yu et al., 2020a). Our recent study (Li et al., 2021) has found a large
variability of $O_3$-precursor relationship in spatiotemporal scales at Zibo, China based
on a July 2019 field campaign, and this phenomenon may occur widely in many other
cities (Lu et al., 2010; Lyu et al., 2016), which challenges current $O_3$ pollution control
(Wang et al., 2017a; Xue et al., 2014b).
**Table 1** summarizes the published studies of $O_3$-precursor relationship using the
0-D box model (implemented with different gas-phase chemical mechanisms) at
diversified patterns of time scale in many places of China. The observational period in
most previous studies was short-term (i.e., less than one month), while medium-term
(i.e., from one to several months), and long-term (i.e., multiple years) periods were
limited. As shown in **Table 1**, we find that model input datasets with different
timescales have been employed in previous studies to identify the campaign-averaging
$O_3$ formation regime, but there is a lack of comparison among these different timescales.
We also find that there are more than half cases using the averaged diurnal patterns as
box model input, which is particularly common for those medium and long-term



measurements. For example, a 10 years long-term observational study by Wang et al.,
(2017a) applied monthly pattern of time scale for model simulation with the reason for
saving computational resources, and it also revealed a substantial temporal variability
of $O_3$-precursor relationship. In addition, it is believed that long-term (measurements
of at least several months) and multiple-site continuous online measurements can
provide opportunity to develop $O_3$ control strategy more comprehensively over a wider
spatiotemporal scale (Li et al., 2021; Wang et al., 2017b; Wang et al., 2017b). However,
such measurements have been quite rare in China, limiting the present understanding
of $O_3$-precursor relationship (Lu et al., 2019; Wang et al., 2017b).
In this study, a five-month field campaign was conducted in the summer of 2019
to investigate the ozone formation chemistry at 3 sites in Zibo, a major prefecture-level
Chinese city in Shandong province. According to our measurements at the three sites in
Zibo, the averaged $O_3$ concentration during the whole observational period was around
50 ppbv, while the daily maximum of $O_3$ concentrations for some extremely polluted
periods were nearly 120-150 ppbv (see details in **Section 3.1**). Here we developed an
approach that integrating multiple patterns of time scale for box model simulation,
which aimed at illustrating the non-linearity of $O_3$-precursor relationship driven by its
actual daily / weekly / monthly variability. Our results can be conducive to interpreting
variations of $O_3$-precursor relationship over a wider spatiotemporal scale, and they
provide implications for developing more precise and constrained $O_3$ control strategies
in other regions.
## 2 Methods
### 2.1 Study sites and measurements

Field measurements were conducted in a major prefecture-level city (Zibo), which
is in the middle of Shandong Province, northern China, from 1 May to 30 September,
2019. **Figure S1** shows the surrounding environment and geographical locations at the
three sampling sites; a detailed description of the Tianzhen (TZ), Beijiao (BJ) and
Xindian (XD) sites can be found in our previous study (Li et al., 2021). Briefly, TZ
contains a mixture of crude oil processing and operation stations and farming areas, and
is classified as suburban area; XD contains a mixture of residential and heavy industrial
zones, and is considered as a suburban area; BJ is in the urban area of Zibo.
Two online gas chromatography–flame ionisation detector (GC-FID, Thermo
Scientific GC5900) systems and one online gas chromatography–flame ionisation
detector/photoionisation detector (GC-FID/PID, Syntech Spectras GC 955-615/815)
system were deployed at TZ, BJ, and XD respectively. These GC systems measured 55
VOC species at a 1-h resolution, and detailed descriptions were given in our previous
study (Li et al., 2021). Typical inorganic gases of $O_3$, NO, $NO_2$, CO and $SO_2$ were
measured using online commercial gas analysers (Thermo Scientific 49i, 42i, 48i and





43i, USA) at the three sites. Meteorological data (i.e., temperature, relative humidity,
UV-A solar radiation, precipitation, wind speed, and wind direction) were continuously
monitored by the Zibo Eco-Environmental Monitoring Center at the three sites.
**Table S1** summarized the limit of detection, accuracy, precision of the instruments
at the three sites, and all the measurement instruments were regularly subjected to the
service of checking and maintenance during the whole campaign. As for VOC
measurement, two online gas chromatography–flame ionisation detector (GC-FID,
Thermo Scientific GC5900) systems were automatically operated with a time resolution
of 1 h at TZ and BJ sites, and measured VOC species were separated into $C_2$-$C_5$ and $C_6$-
$C_{12}$ VOCs. For $C_2$-$C_5$ VOCs, a GC with pre-concentration is used by desorption and
separation on a combination of two columns respectively, then a FID detector is applied
for quantification. For $C_6$-$C_{12}$ VOCs, air samples are pre-concentrated on Tenax GR and
subsequently separated by chromatographic column, then detected by another FID
detector.
Similarly,    one    online    gas    chromatography–flame    ionisation
detector/photoionisation detector (GC-FID/PID, Syntech Spectras GC 955-615/815)
system was deployed with time resolution of 1 h at XD site. For $C_2$-$C_6$ VOCs, the
hydrocarbons are concentrated on a Tenax GR carrier, then thermally desorbed and
separated on a DB-1 column, and finally detected by FID and PID detectors. For $C_6$-
$C_{12}$ VOCs, the air sample is concentrated on a Carbosieves SIII carrier at 5°C, then
thermally desorbed and separated on a system consisting of two columns, and FID and
PID detectors are employed for subsequent detection. More details of online VOC
measurement also can be found elsewhere (Chien, 2007; Jiang et al., 2018; Xie et al.,
2008).

To ensure the quality assurance / quantity control (QA/QC) of online VOC
measurement, two five-point calibrations (i.e., 2, 4, 6, 8, 10 ppbv) for standard gases
with 55 VOC species (Linde Co., Ltd, USA) were carried out in May and August of
2019 at the three sites. **Table S2** showed that the calibration linearity ($R^2$) of all
measured VOCs were nearly 0.9990. Additionally, a single-point calibration (i.e., 6
ppbv) was regularly performed every month during the whole campaign. As shown in
**Figure S2** (a case from TZ), the retention time, peak fitting and baseline of the total ion
current (TIC) chromatogram were manually checked and adjusted on a daily basis.

### 2.2 0-D box model and design of four patterns of time scale

The 0-D box model integrated with the latest Master Chemical Mechanism of
MCMv3.3.1 (http://mcm.york.ac.uk/) has been widely utilized in many regions (He et
al., 2019; Jenkin et al., 2015; Liu et al., 2019; Whalley et al., 2021). Unlike the lumped
chemical mechanisms such as CB05 (Wang et al., 2017a; Yarwood et al., 2005), CB6
(Yarwood et al., 2010), RACM/RACM2 (Goliff et al., 2013; Stockwell et al., 1997,


2020) and SAPRC-07 (Carter, 2010), the MCMv3.3.1 is a near-explicit chemical
mechanism consisting of over 5,800 species and 17,000 reactions (Jenkin et al., 2015;
Saunders et al., 2003), which can be used to describe the gas-phase chemistry (i.e., in-
situ photochemistry). In this study, the box model (F0AM) (Wolfe et al., 2016) was
applied and constrained by the mean diurnal profiles of meteorological data (i.e.,
temperature, relative humidity, and photolysis rates), 4 inorganic gases (i.e., $SO_2$, CO,
NO, and $NO_2$), and 45 speciated VOCs (in the VOC species list of MCMv3.3.1; see
**Table S3**). Since measured photolysis rates ($J$ values) were not available, the measured
UV-A solar radiation was used to scale the photolysis rates calculated from the
Tropospheric    Ultraviolet    and    Visible    Radiation    model    (TUVv5.2;
https://www.acom.ucar.edu/Models/TUV/Interactive_TUV/) following the approach of
recent studies (Lyu et al., 2019; Lyu et al., 2016). A dilution rate of $3/86400$ s$^{-1}$ was
applied for all non-constraint species and simulation days through a stepwise sensitivity
test by adjusting it from $1/86400$ s$^{-1}$ to $5/86400$ s$^{-1}$ (see details in **Text S3**). For each
model run (i.e., each daily model simulation), it was performed on a daily basis with
intervals of 24 hours spanning from 0:00 to 23:00, and each individual model simulation
was run to reach one-day diurnal steady state. The detailed descriptions of box model
operation were provided in our previous study (Li et al., 2021).
Since the box model simulations are conducted with intervals of 24 hours spanning
from 0:00 to 23:00 local standard time (Wang et al., 2018), the entire campaign
observations were taken into four patterns of time scale (i.e., five-month, monthly,
weekly, and daily) as diurnal average format for model input (**Figure 1**). Note that some
days or weeks were not modeled due to significant miss in the measurements.
Nevertheless, the total simulation number at the daily (i.e., 100, 81, and 114 days for
TZ, BJ and XD respectively) or weekly (i.e., 21, 20, and 19 weeks for TZ, BJ, and XD
respectively) scale was representative of the five-month campaign. Specifically, the
entire campaign dataset was processed into four patterns of time scale, and were
modeled as base runs. Then we performed the sensitivity modeling to calculate the
relative incremental reactivity (RIR) of precursors by adjusting the input concentrations
in the base runs (see next section) (Lu et al., 2010a).

**2.3 Calculation of net $O_x$ production rate $P(O_x)$ and Relative incremental
reactivity (RIR)**

Considering the rapid chemical titration of NO to $NO_2$ in the presence of $O_3$, the
concept of 'total oxidant' ($O_x = O_3 + NO_2$) has been widely used to represent the actual
photochemical production of $O_3$ (Lu et al., 2010). Similar to those described in previous
studies using the 0-D box model (He et al., 2019; Lyu et al., 2016), the net or in-situ $O_x$
production rate ($P(O_x)$) is defined as the difference between the $O_x$ gross production
rate ($G(O_x)$) and the $O_x$ destruction rate ($D(O_x)$), which is formulated in accordance



with Eq. (1):
$$P(O_x) = G(O_x) - D(O_x) \tag{1}$$
The $O_x$ gross production rate ($G(O_x)$), or the total chemical production of $O_x$, is
calculated by summing the rates of oxidation of NO by $HO_2$ and $RO_2$ radicals in
accordance with Eq. (2):
$$G(O_x) = k_{HO_2+NO}[HO_2][NO] + \sum k_{RO_{2,i}+NO}[RO_{2,i}][NO] \tag{2}$$
The $O_x$ destruction rate ($D(O_x)$), or total chemical loss of $O_x$, is calculated by
summing $O_3$ photolysis, the reaction of $O_3$ with OH, $HO_2$ and alkenes, as well as the
reaction between $NO_2$ and OH, as described by Eq. (3):
$$D(O_x) = k_{O^1D+H_2O}[O^1D][H_2O] + k_{OH+O_3}[OH][O_3] + k_{HO_2+O_3}[HO_2][O_3] +$$
$$k_{alkenes+O_3}[alkenes][O_3] + k_{OH+NO_2}[OH][NO_2] \tag{3}$$
Concentrations of radicals and intermediates are obtained from the outputs of the
0-D box model. The $k$ values in Eq. (2) and (3) represent the rate constants of the
corresponding reactions, respectively. The subscript '$i$' in Eq. (2) represents the
individual $RO_2$ species.
Additionally, relative incremental reactivity (RIR) has been widely used as a
metric to quantify the $O_3$-precursor relationship, and it can be derived from the 0-D box
model (MCMv3.3.1) by changing the input mixing ratios of its precursors (Sillman,
2010; Xue et al., 2014a). The RIR is defined as the ratio of percentage change in net $O_x$
($O_x = O_3 + NO_2$) production rate $P(O_x)$ (Li et al., 2021) to percentage change of
concentration of precursor X. The RIR of a specific precursor X is described in Eq. (4):
$$RIR(X) = \frac{[PO_x(X) - PO_x(X - \Delta X)]/PO_x(X)}{\Delta C(X)/C(X)} \tag{4}$$
Here, X is a specific precursor (i.e., $NO_x$, CO or grouped / individual VOC species),
C(X) is the measured concentration of precursor X, and $\Delta C(X)$ is the concentration
change caused by the hypothetical change ($\Delta X$, 10% of X in this study in accordance
with the previous studies (Lyu et al., 2016; Wang et al., 2018)) in precursor X. Therefore,
$\Delta C(X)/C(X)$ was 10% in this study. $PO_x(X)$ represents the simulated $O_x$ production rate
in a base run, whereas $PO_x(X-\Delta X)$ is the simulated $O_x$ production in a second run with
a hypothetical concentration change (10%) of species X. Obviously, a higher positive
value of RIR(X) suggests a more effective way of reducing the ambient $O_3$ production
rate by reducing X (Ling et al., 2011; Zhang et al., 2008a). In this study, the RIR method
was applied mainly to evaluate the $O_3$-$NO_x$-VOC sensitivity and determine the
photochemical regimes among four patterns of time scale.



## 3 Results and discussion

### 3.1 Overview of the field campaign

**Figure 2** shows the time series of measured meteorological parameters and $O_3$ as well as its precursors at the three sites during the whole campaign. In general, the temperature ($T$) and relative humidity (RH) were basically consistent at the three sites, while the wind speeds were different, which suggests that the three sites had an overall consistent meteorological condition. In addition, the time series of UV-A radiation was shown in **Figure 2d**, which was only available from one urban site of Zibo but expected to represent the whole Zibo city in this study. Following the protocol of the previous studies (Lyu et al., 2019; Wang et al., 2017b; Xue et al., 2014), the time series of photolysis rates (e.g., $J_{NO_2}$ (**Figure 2e**) and $J_{O^1D}$ (**Figure 2f**)) were calculated from TUVv5.2 model and further scaled from UV-A radiation measurement.

As shown in **Figure 2g**, we found that severe $O_3$ pollution was observed at the three sites throughout the whole campaign. According to our measurements at the three sites in Zibo, the averaged $O_3$ concentration during the whole observational period was around 50 ppbv, while the daily maximum of $O_3$ concentrations for some extremely polluted periods were nearly 120-150 ppbv (**Figure 2g**). Interestingly, the $O_3$ concentrations at the three sites were generally consistent, while the levels of its precursors (e.g., VOC, NOx) were obviously different (**Figure 2h-k**), which implies the site-to-site variation of $O_3$ formation chemistry for the whole Zibo city.

Generally, OH reactivity (or OH loss rate, $k_{OH}$) is widely applied to quantity the capacity of OH consumption by VOCs (Tan et al., 2019a). According to **Table S3**, the BVOC reactivity ($k_{BVOC}$, $3.5 \pm 4.1$ $s^{-1}$) in TZ were highest within the three sites. As BJ was mainly influenced by the emission from urban region, it showed the highest AVOC reactivity ($k_{AVOC}$, $6.8 \pm 6.3$ $s^{-1}$) and NOx level ($31.1 \pm 28.6$ ppbv). In addition, XD showed the highest level of alkenes* reactivity (anthropogenic alkenes which excludes isoprene in this study) of $4.0 \pm 3.2$ $s^{-1}$ within the three sites, and the local petrochemical industry nearby XD area may explain such characteristic (Li et al., 2021).

### 3.2 Evaluation of box model performance

The measured $O_3$ concentrations were not constrained in our MCMv3.3.1 box model calculation, thus the model performance could be quantitatively assessed by comparing the modeled $O_3$ (from base runs) with the measured $O_3$. **Figure S3-S8** show the time series of simulated and observed $O_3$ concentrations at four patterns of time scale. In most cases, the box model simulation could accurately capture the level and variation trend of the observed $O_3$. However, on some days the modeling results underestimated or overestimated the $O_3$ concentrations. Such discrepancies between the simulated and observed $O_3$ were likely due to limitations in explicit representations of



atmospheric and transport processes (i.e., the horizontal and vertical transport process
of ground ozone) by 0-D modeling approach (Lyu et al., 2019; Yu et al., 2020b).
Specifically, ozone simulated by the 0-D box model is considered as in-situ
photochemical processes from its precursors. Unlike the 3-D air quality model, 0-D box
model usually simplifies the representation of the physical processes (i.e., deposition
and advection) (Lu et al., 2010a; Sillman, 2010). Note that some adjustable parameters
(e.g., radiation scheme, dilution rate) were remained consistent in all of our model
calculations, which ensured the comparability of model results to the greatest extent.
The index of agreement ($IOA$) (Li et al., 2021; Lyu et al., 2016), Pearson's
correlation coefficient ($r$) and root mean square error ($RMSE$) were jointly used as
statistical metrics to quantify the goodness-of-fit between the simulated and observed
$O_3$ concentrations. **Table S4** summarizes these statistical metrics for each site at various
patterns of time scale. Because any single statistical metric has its own limitations,
using these three indicators conjointly provided a more comprehensive evaluation of
the model performance (Su et al., 2018b). Generally, higher $IOA$ and $r$ as well as lower
$RMSE$ indicate better agreement between the simulated and observed values (Wang et
al., 2018; Willmott, 1982). As shown in **Table S4**, slightly reduced correlation was
observed as the time scale changed from the wider (i.e., five-month scale) to the
narrower (i.e., daily scale) pattern, which is understandable because of the enlarged
statistical samples in the narrower pattern of time scale.
In summary, TZ showed the best performance of the box model simulation,
followed by XD and BJ, regardless of any statistical metrics or different patterns of time
scale. The overall model performance in this study (i.e., a day-to-day $IOA$ of
approximately 0.90 for TZ) was close to or slightly better than those reported in
previous studies, such as $IOA = 0.74$ in Hong Kong (Liu et al., 2019), $IOA = 0.74$ in
Wuhan (Lyu et al., 2016) and $IOA = 0.90$ in Jiangmen (He et al., 2019). According to
the above evaluation of base runs, our modeled results were acceptable for the
subsequent $O_3$-precursor relationship analysis described in the following sections.
**3.3 Month-to-month**
The $O_3$ precursors were divided into four major categories, including
anthropogenic VOC (AVOC), biogenic VOC (BVOC, only isoprene in this study), CO
and $NO_x$ (Tan et al., 2019b). AVOC was further divided into three subcategories: alkanes,
aromatics and alkenes* (the asterisk denotes anthropogenic alkenes, excluding isoprene
in this study) (Yu et al., 2020a). Additionally, the RIR values of major precursor groups
(i.e., AVOC, BVOC, CO, $NO_x$, alkanes, alkenes* and aromatics) were calculated to
further quantify the $O_3$-precursor relationship (see section 2.3 for more details). **Figure**
**3a-b** presents the monthly RIR values of the major precursor groups at each site, and
the large variability of $O_3$-precursor relationship at spatiotemporal scale (i.e., site-to-



site and month-to-month) was observed. Specifically, in most months, XD generally
showed the highest $RIR_{AVOC}$ among the three sites, followed by BJ and TZ. In addition,
$RIR_{BVOC}$ showed similar level to $RIR_{AVOC}$ in TZ, but much less than $RIR_{AVOC}$ in BJ and
XD, which can be explained by the observed higher BVOC reactivity in TZ than the
other two sites (see **Figure S13** and **Table S3**). Also, almost all the precursor groups
showed positive RIR values, except negative $RIR_{NOx}$ appeared in BJ and XD in
September. Among the three subcategories of AVOC, alkenes* always had the highest
RIR values, followed by aromatics, while the contribution of alkanes to $O_3$ formation
can be ignored due to their near-zero RIR values. That sequence of $O_3$-AVOC sensitivity
(alkenes* > aromatics > alkanes) indicated by the RIR analysis was consistent with
previous studies in some other Chinese cities (Su et al., 2018b; Tan et al., 2019b).
Significant monthly variations of $O_3$, NOx, CO, VOC reactivity and TVOC/NOx ratios
(in ppbC/ppbv, as a widely used simple metric to determine the photochemical regime)
(National Research Council, 1991) were also observed from May to September (see
**Figure S13** and **Table S3**) at the three sites, which indicates the temporal variation of
local primary emission for $O_3$ precursors.

$O_3$ formation chemistry is usually classified into two regimes (i.e., VOC-limited
and NOx-limited) or three regimes (i.e., VOC-limited, transitional and NOx-limited)
(He et al., 2019; Wang et al., 2018). In this study, $RIR_{NOx}/RIR_{AVOC}$ (the ratio of two RIR
values) was used as a metric to classify the photochemical regimes (Li et al., 2021).
Specifically, $RIR_{NOx}/RIR_{AVOC}$ value of less than 0.5 was defined as VOC-limited regime,
greater than 2 as NOx-limited regime, and from 0.5 to 2 as transitional regime (see **Text
S2** and **Table S5**) (Li et al., 2021). **Figure 3c** shows monthly $RIR_{NOx}/RIR_{AVOC}$ at each
site, which clearly reveals the spatial and temporal variations in photochemical regimes.
For instance, the photochemical regime at the TZ site was considered to be transitional
regime in May, NOx-limited regime in June and July, and VOC-limited regime in
August and September; whereas for a specific month like June, NOx-limited, VOC-
limited, and transitional regimes were generally identified for TZ, BJ, and XD
respectively. **Figure 5b** shows good consistency between monthly TVOC/NOx and
$RIR_{NOx}/RIR_{AVOC}$, suggesting that the changes of local emissions for $O_3$ precursors may
explain the considerable variation of $O_3$ formation chemistry in different months.
**3.4 Week-to-week**

**Figure 4** shows the time series of week-to-week RIR values of major precursor
groups and $RIR_{NOx}/RIR_{AVOC}$ at three sites in Zibo. Compared with month-to-month
results, **Figure 4** further reveals the $O_3$-precursor relationship with more information in
temporal trends. The temporal variations in weekly $RIR_{AVOC}$ at the three sites generally
decreased and then increased, whereas weekly $RIR_{NOx}$ represented an opposite temporal
variation during the entire campaign. Additionally, weekly $RIR_{BVOC}$ showed a trend of


first decrease and then increase at TZ, while it did not show clear temporal variation at
BJ and XD due to low values (**Figure 4a-c**). In general, $RIR_{alkanes}$, $RIR_{alkenes*}$ and
$RIR_{aromatics}$ showed a tendency consistent with that of the $RIR_{AVOC}$ at three sites (**Figure**
**4d-f**). Overall, these phenomena were consistent among the three sites, though the
magnitude of RIR values varied site-to-site. In parallel, the temporal changing of $O_3$
precursor (e.g., AVOC, NOx) was also observed at the three sites during the entire
campaign (see **Figure S14**). For example, the weekly NOx concentration showed an
overall trend of first decrease and then increase, while the AVOC reactivity showed a
different temporal variation. Given the good consistency between weekly TVOC/NOx
and $RIR_{AVOC}/RIR_{NOx}$ (**Figure 5c**), the temporal variations of RIR values and $O_3$
formation chemistry at the three sites may be elucidated by the emission changes of $O_3$
precursors.

As shown in **Figure 4g-i**, all the three sites showed similar temporal trends of
$RIR_{NOx}/RIR_{AVOC}$, as it increased first and then decreased, though the magnitude of
$RIR_{NOx}/RIR_{AVOC}$ varied largely at each site. Such site-to-site variability of
$RIR_{NOx}/RIR_{AVOC}$ suggests that the photochemical regime in a local scale was mainly
influenced by local emissions. By contrast, the site-to-site synchronization in temporal
trend of $RIR_{NOx}/RIR_{AVOC}$ suggests that the photochemical regime in a local scale may
also be influenced by the emissions in a regional area. Therefore, the long-term, week-
to-week $RIR_{NOx}/RIR_{AVOC}$ of multiple sites can further reflect the variability of ozone
formation regime at a large geographic scale.
**3.5 Day-to-day**

In this section, $O_3$-precursor relationship at the narrowest pattern of time scale was
identified in detail. **Figure S9-S10** shows the time series of daily RIR values at three
sites in Zibo, where the temporal trend of RIR values was consistent with that at weekly
scale (**Figure 4**). Additionally, the time series of daily $RIR_{NOx}/RIR_{AVOC}$ (**Figure S11**)
first increased and then decreased during the entire campaign, which was also consistent
with that of weekly scale. In summary, the time series of RIR values from the daily
scale can provide more informative variations and characteristics of $O_3$-precursor
relationship in temporal trends.

**Table 2** summarizes the number of days and proportions that were classified into
the three photochemical regimes across each site and each pattern of time scale. Near-
consistent proportions of $O_3$ formation regimes (using $RIR_{NOx}/RIR_{AVOC}$ as a metric)
were shown among multiple patterns of time scale, whereas a variability of proportion
occurred among the three sites. The proportions of photochemical regimes changed
accordingly along with the time scale varied from wider to narrower pattern. Taking TZ
as an example, 20% (monthly) and 26% (daily) of the time was considered as VOC-
limited regime. The number of days and proportions for photochemical regimes



summarized at four patterns of time scales can reveal a more plausible and
comprehensive variation in ozone formation chemistry. Compared with patterns of
monthly and weekly scales, the results derived at a daily scale can reveal the temporal
variability of photochemical regimes more comprehensively. Note that the
photochemical regime proportion obtained from the day-to-day scale has an advantage
due to the large number of statistical samples.

**3.6 Comparison among different patterns of time scale**

This section gives a more comprehensive understanding of the campaign-
averaging $O_3$-precursor relationship by comparing the similarities and differences of
the results from various patterns of time scale. The overall $O_3$-precursor relationship for
the entire campaign can be quantified by averaging the RIR values from the individual
simulation runs depending on the chosen time scale (e.g., five simulation runs for
monthly scale in this study). Therefore, four sets of logical and comparable results can
be derived to represent the campaign-averaging $O_3$-precursor relationship, as four
patterns of time scale (i.e., five-month, monthly, weekly, and daily) were treated in this
study.
**Figure 6** shows the averaged RIR values of the major precursor groups at different
patterns of time scale. As the time scale changed from wider (i.e., five-month scale) to
narrower (i.e., daily scale) pattern, all three sites showed increased $RIR_{AVOC}$ and
$RIR_{alkenes*}$ as well as decreased $RIR_{NOx}$, whereas the RIR of other precursors (i.e.,
BVOC, CO, alkanes and aromatics) did not vary obviously (see **Table S6**). Comparing
the $O_3$-VOC-$NO_x$ sensitivity at the daily scale, the results obtained at the five-month
scale underestimated $O_3$-AVOC sensitivity by 48% (TZ), 66% (BJ), and 49% (XD), and
overestimated $O_3$-$NO_x$ sensitivity by 37% (TZ), 142% (BJ), and 144% (XD). We
performed comprehensive uncertainty analysis for model input and output results,
which was assessed through statistical methods (see details in **Section 3.7**). We found
that the model-derived RIR values may become more uncertain when the input dataset
was averaged into a wider diurnal pattern (i.e., five-month scale), which may explain
the discrepancy of RIR values between five-month scale and daily scale. We expect that
such discrepancies derived from different patterns of time scale could widely exist in
many other world areas. Note that the mean RIR values were generally consistent
among the four patterns of time scale within a reasonable range (within 25-75[th] quantile
and standard deviation, see **Figure 6** and **Table S5**), suggesting that any selected pattern
of time scale could reasonably derive the campaign-averaging $O_3$-precursor relationship.
**Figure 7** further shows the variations in photochemical regimes (defined by
$RIR_{NOx}/RIR_{AVOC}$; see **Text S2** and **Table S5** for details) for each pattern of time scale.
Specifically, TZ was mainly considered as transitional regime for the entire campaign
period, whereas its variations covered three photochemical regimes, which was





consistent with the results from **Table S6**. BJ was generally identified as VOC-limited
regime, whereas some days were also grouped into transitional regime. XD was
considered as primarily between VOC-limited and transitional regime, and its
variations also spanned three photochemical regimes. Compared with the five-month
pattern, it was further found that the averaged $RIR_{NOx}/RIR_{AVOC}$ from other time scale
patterns (i.e., monthly, weekly, and daily) were higher (12% to 20% for TZ; 38% to
153% for XD) or lower (21% to 65% for BJ) than that from five-month scale. Note that
the above discrepancies in photochemical regime derived from multiple patterns of time
scale may influence the development of targeted $O_3$ control strategies. In summary, the
photochemical regime derived by averaging $RIR_{NOx}/RIR_{AVOC}$ from the daily scale (see
**Table S6**) suggests that the three sites mainly followed the sequence of TZ (1.34 ±
1.39) > XD (0.67 ± 1.49) > BJ (0.16 ± 0.65).
In addition, the temporal variations of TVOC/NOx in different timescales were
identified during the whole campaign, and good correlations between observed
TVOC/NOx and model derived $RIR_{NOx}/RIR_{AVOC}$ at four patterns of time scale were
also found (see **Figure 5**). Such consistency suggests that both metrics can reasonably
reflect the variation of photochemical regimes, which can also improve the reliability
of our box model simulation.
The consistency and difference of model output (summarized in **Table S7**) are
quantified by the statistical methods of Pearson's correlation coefficient (Hu et al., 2018)
and paired-samples *t*-test analysis (Wang et al., 2016). In particular, we assess and
compare the degree of significance of differences among multiple patterns of time scale
by the *p* values (a statistical significance assuming at $p < 0.05$) through paired-samples
*t*-test and Wilcoxon matched-paired signed-rank test (non-parametric statistics)
(Chiclana et al., 2013). **Figure 8a** shows high Pearson's correlation coefficients (with
values all above 0.85, $p < 0.01$) were found among four patterns of time scale, and the
higher correlation coefficient was identified between the two closer patterns. **Figure
8b-c** shows that the differences among multiple patterns of time scale were non-
significant using Paired-samples t-test analysis and Wilcoxon matched-pair signed-rank
test respectively. Furthermore, their results indicate that more significant difference was
recognized between the two distant patterns (e.g., daily and five-month), which is
consistent with the results of Pearson's correlation analysis. Noted that the discrepancy
between the two distant patterns was not significant but non-negligible (e.g., $p = 0.092$
of Wilcoxon matched-paired signed-rank test between five-month and daily patterns).
The influence of different patterns of time scale on deriving RIR values from
individual AVOC species was further investigated. Briefly, quantifying the relative
contribution of individual AVOC on $O_3$ formation based on RIR calculation is beneficial
to the development of cost-effective AVOC control strategies (Zhang et al., 2021).
**Figure 9** shows the averaged RIR values of individual AVOC species (i.e., top 10) at



different patterns of time scale (i.e., five-month, month-to-month, week-to-week) at
three sites in Zibo. As shown in **Figure 9**, the 10 individual AVOC species at the three
sites were selected according to the top 10 highest RIR from five-month pattern. All
three sites showed that the RIR of individual AVOC species increased gradually as the
time scale changed from the wider (i.e., five-month) to narrower (i.e., weekly) pattern,
which was consistent with the earlier discussion (see **Figure 6** and **Table S6**) of $O_3$-
AVOC sensitivity derived from four patterns of time scale. The results also indicate that
the choice of time scale pattern has a limited effect on deriving high-ranking AVOC
species (i.e., top 10) based on RIR calculations.
**3.7 Uncertainty analysis**

The uncertainty of model input comprehensively was assessed and quantified in
this section, which is embedded in pre-processed dataset with multiple patterns of time
scale. The box model simulation was performed on a daily basis with intervals of 24
hours spanning from 0:00 to 23:00 local standard time. As showed in **Figure 1**, the daily
simulation used the individual daily pattern to constrain model, while the input dataset
of averaged diurnal patterns (i.e., weekly, monthly, and five-month) is treated by
averaging individual daily pattern into different timescales. Note that compared with
the model input data from daily pattern, the discrepancies of $O_3$ precursor groups from
averaged diurnal patterns (i.e., weekly, monthly, and five-month) were overall limited
(see **Table S7**), which is reasonably for such kind of time series observation data.
However, as shown in our analysis in previous section, the model input dataset with
averaged diurnal patterns (i.e., weekly, monthly, and five-month) will result in non-
negligible discrepancy and uncertainty of model output. Therefore, the standard
deviation of averaged diurnal patterns was employed to quantify the uncertainty of
model input dataset. **Figure 10** shows the distributions of the standard deviations for
OH reactivity ($k_{OH}$) or concentration of $O_3$ precursor groups at three averaged patterns
of time scale at the three sites. As the time scale changed from wider (i.e., five-month
scale) to narrower (i.e., weekly scale) pattern, the uncertainty (indicated by the average,
median and 25%-75% quantile of the standard deviations) decreased accordingly. Note
that the 10-90% quantile of the standard deviations partly presented with different trend,
which was due to the enlarged statistical samples at narrower (i.e., weekly scale) pattern.
Generally, such uncertainty of model input dataset will lead to the discrepancy of model
output results, especially for determining $O_3$ formation chemistry at the wider pattern
of time scale.

Moreover, it has been widely recognized that the uncertainty for 0-D box model
simulation mainly arises from the constraint of observation dataset and the
configuration of model scheme. Note that constraints with more species from
measurements (or including as many species as possible) would lower its uncertainty





from the chemical box model simulation (Wolfe et al., 2011, 2016). Nevertheless, due
to the measurement limitation in our field campaign, we are unable to measure some
important atmospheric species (i.e., HONO and oxygenated VOC (OVOC)), and these
may arise uncertainty in box model simulation. For instance, Xue et al., (2021)
performed a sensitivity test for HONO constraint in their box model simulation, and
they showed that without HONO constraint would lead to $O_3$ photochemical production
rate decreasing by 42%. More recently, Wang et al., (2022) obtained a comprehensive
VOC dataset at Guangzhou, and their results showed that box model simulation without
OVOCs constraints would underestimate the productions of ROx and $O_3$. In addition,
the parameter configuration of model scheme is essential to derive a reliable and valid
model output. Dilution rate is an important model technical parameter, which is
essential to obtain a reliable model output result. We performed a stepwise sensitivity
test for this parameter to obtain an optimized dilution rate, and assigned it to all non-
constraint species, which can reduce uncertainty in box model simulation (see details
in **Text S3**). Also, the dry and/or wet deposition of pollutants is an important
atmospheric physical process, which has been mostly parameterized in emission-based
chemical transport modeling but very limited in box model, as most of the primarily
emitted species are already constrained from measurements. Xue et al., (2014)
considered $O_3$ deposition into box model simulation, and their result showed negligible
contribution of $O_3$ deposition to total $O_3$ destruction rates. As for this work, we are
unable to consider the deposition due to the difficulty in representing and
parameterizing this term in the 0-D box model. Nevertheless, deposition of $O_3$ and other
species may be one of the uncertainties during box model simulation, which is worth
further study in the future.
**4 Summary and implications**
Our present results suggest that comprehensively understanding of multiple
patterns of time scale is conductive to formulating a more accurate and robust $O_3$ control
strategy. Specifically, as identified from the narrower patterns of time scale (i.e., weekly
and daily), the site-to-site photochemical regime indicated by $RIR_{NOx}/RIR_{AVOC}$ showed
various magnitudes but a synchronous temporal trend. This indicates that the $O_3$
formation regime in a city area can be influenced by local and regional emissions jointly.
The reason behind this phenomenon is not clear at present, and we believe that further
investigation on the synergetic effect of local and regional emission reduction for $O_3$
control would help elucidating this observation. It was also found that the campaign-
averaging photochemical regimes showed overall consistency but non-negligible
variability among the four patterns of time scale, which was mainly due to the
embedded uncertainty in model input dataset with averaged diurnal patterns. This
implies that comparison among multiple patterns of time scale based on RIR analysis





is useful to derive the $O_3$-precursor relationship more accurately and reliably.
Moreover, the high-ranking AVOC species (i.e., top 10) based on RIR calculations
were overall consistent from the narrow to wide patterns of time scale. This
demonstrates that datasets with wider pattern of time scale can still produce an accurate
RIR ranking / prioritization for VOC control. **Table S8** summarizes the total run number
of box model for different patterns of time scale. It is known that large-scale computing
capacity and computational efficiency were required in the narrower pattern of time
scale (e.g., 2760 simulation runs in weekly scale in this study). Considering the
difficulties of performing long-term and continuous online measurements in some
environments, it is also advisable to identify the high-ranking VOC species from the
campaign-averaging diurnal pattern in box model simulation.
In this study, we explored the non-linearity of $O_3$-precursor relationship in a way
driven by the actual daily / weekly / monthly variability around the distribution. Our
results highlight the importance to quantitatively test the impact of different timescales
on photochemical regime determination, as there is uncertainty embedded in model
input dataset when averaging individual daily pattern into different timescales. Such
understanding would be complementary in developing more accurate $O_3$ pollution
control strategy, particularly as the long-term $O_3$-precursor observations (e.g., from
several months to years) are becoming more available than before in many places of
China. In addition, site-to-site difference of model-derived photochemical regimes also
underlines the importance of developing target $O_3$ control strategy for different areas in
a city scale. Specifically, according to the averaged $RIR_{NOx}/RIR_{AVOC}$ at daily pattern,
the derived photochemical regime was transitional for TZ (suburban) and XD
(suburban), while VOC-limited for BJ (urban). This implies that for mitigating ozone
pollution in Zibo city, more endeavors should be devoted to the anthropogenic VOC
reduction in urban areas, while strengthening the synergetic mitigation of VOC and
NOx emissions at the same time in other suburban areas. Although the above
implications for $O_3$ control were derived from a case study in a major prefecture-level
city (Zibo) of northern China, the developed approach by integrating multiple patterns
of time scale in the present work can be used to other regions, particularly the on-going
"One City One Policy" campaign (2021-2023) for $O_3$ control in many cities in China.



## Acknowledgement

This work was supported by National Center for Air Pollution Prevention and Control
(No. DQGG202119) and Ministry of Science and Technology PRC (No.
G20200160001). We also thank Prof. William Bloss for helpful comments.

## Data and code availability

The code for the Master Chemical Mechanism (MCMv3.3.1) can be achieved from
http://mcm.york.ac.uk/. The datasets generated during and/or analysed during the
current study are available from the corresponding author on reasonable request.

## Author contribution

KL conceived the study; ZZ performed the modeling; ZZ, KL, and ZB analyzed the
data; BX, JD, LL, SL, CG, and WY conducted the field measurement; ZZ and KL wrote
the paper with assistance of interpretation and revision from all authors. All authors
contributed to the manuscript preparation and discussions.

## Conflicts of interest

The authors declare that they have no conflicts of interest.

## Supplement

The supplementary discussion of RIR calculation of different hypothetical changes,
determining the photochemical regime, sensitivity test of different dilution rates, and
detailed box modeling results are provided in **Text S1-S3**, **Table S1-S8** and **Figure S1-**
**S18**.





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



**Table 1.** Summary of published 0-D box model studies in China

| City | Site | Type | Period | Patterns of Time scale[a] | Mechanism | Reference |
|---|---|---|---|---|---|---|
| Beijing | PKU[b] | Urban | 10 Aug–10 Sep 2006 | Day-to-day (25 d) | CB-IV | (Lu et al., 2010) |
| | YUFA | Suburban | 13–29 Apr 2015, 11–29 Aug 2015, 22 Feb–12 Mar 2016 | Entire period | RACM2 | (Qin et al., 2018) |
| | PKU | Urban | 2–19 Jul 2014 | Entire period | RACM2 | (Tan et al., 2019b) |
| | Beijing | Urban | 1 Jun–6 Jul 2013 | Day-to-day (2 d) | MCMv3.3.1 | (Zong et al., 2018) |
| Dezhou | Yucheng | Rural | | | | |
| Shenzhen | SYY[c] | Urban | 28 Sep–31 Oct 2018 | Entire period | RACM2 | (Yu et al., 2020b) |
| | Fucheng | Urban | | | | |
| Hong Kong | TC | Suburban | 10 Aug–21 Oct 2013 | Entire period | MCMv3.2 | (Zeng et al., 2018) |
| | Wan Shan | Island | | | | |
| | Tung Chung | Urban | Sep-Nov 2002, 2007, 2012 | Year-to-year (3 yrs) | MCMv3.2 | (Xue et al., 2014b) |
| | Qing Sha | Urban | | | | |
| | Tai O | Urban | 23 Oct–1 Nov 2007 | Day-to-day (10 d) | CB-IV | (Cheng et al., 2010) |
| | Tung Chung | Urban | Jan 2005-Dec 2014 | Month-to-month (5 months) | CB05 | (Whalley et al., 2021b) |
| Chengdu | Pengzhou | Suburban | 3 Sep–2 Oct 2016 | Entire period | RACM2 | (Tan et al., 2018b) |
| | Pixian | Suburban | | | | |
| | Shuangliu | Suburban | | | | |
| | Chengzhong | Urban | | | | |
| Zhuhai | Qi'ao | Mountain | 25 Sep–28 Oct 2016 | Entire period | MCMv3.2 | (Liu et al., 2021b) |
| Wuhan | HPEMC[d] | Urban | Feb 2013-Oct 2014 | Month-to-month (21 months) | MCMv3.2 | (Lyu et al., 2016) |
| Guangzhou | GZ | Urban | 5–17 Jul 2006 | Day-to-day (16 d) | CB-IV | (Lu et al., 2010) |
| | BZ | Suburban | | | | |
| | Guangzhou | Urban | | | | |
| | Xinken | Nonurban | 4 Oct–5 Nov 2004 | Entire period | SAPRC | (Zhang et al., 2008b) |
| Hangzhou | Zhaohui | Urban | 17 May, 26 Jun 20, Jul 24, Aug 26 Sep | Entire period (5 d) | MCMv3.3.1 | (Zhao et al., 2020) |
| | Xiasha | Suburban | | | | |






| | Site | Type | Period | Time scale | Mechanism | Reference |
|---|---|---|---|---|---|---|
| Huapu | | Urban | | | | |
| Nanjing | NUIST[e] | Suburban | 3 Jul–1 Aug 2018 | Entire period | CB-IV | (Fan et al., 2021) |
| Nanjing | SORPES | Suburban | 22 Sep–7 Oct 2014 | Day-to-day (8 d) | MCMv3.3.1 | (Xu et al., 2017) |
| Yulin | EMB[f] | Urban | 7 Jul–10 Aug 2019 | Day-to-day (13 d) | MCMv3.3.1 | (Yin et al., 2021) |
| Lanzhou | Renshoushan Park | Urban | 19 Jun–16 Jul 2006 | Day-to-day (3 d) | MCMv3.2 | (Xue et al., 2014) |
| Baoding | EPB[g] | Urban | 10–30 Sep 2015 | Day-to-day (5 d) | MCMv3.3.1 | (Wang et al., 2020a) |
| Chongqing | Nan Quan | Suburban | | | | |
| Chongqing | Chao Zhan | Urban | 24 Aug–22 Sep 2015 | Day-to-day (7 d) | MCMv3.2 | (Li et al., 2018) |
| Chongqing | Jin Yun Shan | Urban | | | | |
| Shanghai | Pudong | Urban | 1–31 Jul 2017 | Day-to-day (16 d) | CB-IV | (Lin et al., 2020) |
| Shanghai | Dianshanhu | Suburban | | | | |
| South China Sea | Wanshan | Island | 11 Sep–21 Nov 2013 | Entire period | MCMv3.2 | (Wang et al., 2018) |

[a]Number of days for modeling the patterns of time scale denotes that which was simulated by the box model.
[b]Peking University
[c]Shenzhen Yanjiusheng Yuan
[d]Hubei Provincial Environmental Monitoring Center
[e]Nanjing University of Information Science & Technology
[f]Environmental Monitoring Building
[g]Environmental Protection Bureau





**Table 2.** Summary of the number of days (for model calculation) and proportions that
were classified into the three photochemical regimes across each site and multiple
patterns of time scale.

| Patterns of Time scale | Site | Photochemical regime: $RIR_{NOx}/RIR_{AVOC}$ | | | | | |
|---|---|---|---|---|---|---|---|
| | | $NO_x$-limited: >2 | | Transition: 0.5~2 | | VOC-limited: <0.5 | |
| | | No. of days | Proportion | No. of days | Proportion | No. of days | Proportion |
| Month-to-month | TZ | 2 | 40% | 2 | 40% | 1 | 20% |
| | BJ | 0 | 0% | 3 | 60% | 2 | 40% |
| | XD | 0 | 0% | 2 | 40% | 3 | 60% |
| Week-to-week | TZ | 7 | 33% | 8 | 38% | 6 | 29% |
| | BJ | 0 | 0% | 10 | 50% | 10 | 50% |
| | XD | 3 | 16% | 6 | 32% | 10 | 53% |
| Day-to-day | TZ | 29 | 29% | 45 | 45% | 26 | 26% |
| | BJ | 0 | 0% | 21 | 26% | 60 | 74% |
| | XD | 20 | 18% | 23 | 20% | 71 | 62% |


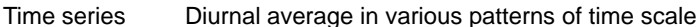

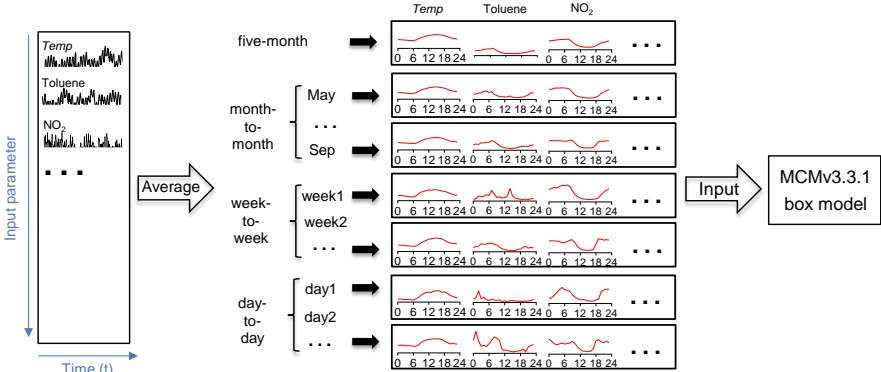

**Figure 1.** Schematic diagram of the dataset treatment to derive four patterns of time scale for 0-D box
model input. Note that the four patterns (i.e., five-month, monthly, weekly, and daily) were the diurnal
average of the initial dataset. This diagram takes one site and several input measurements (temperature,
toluene, and $NO_2$) as examples.



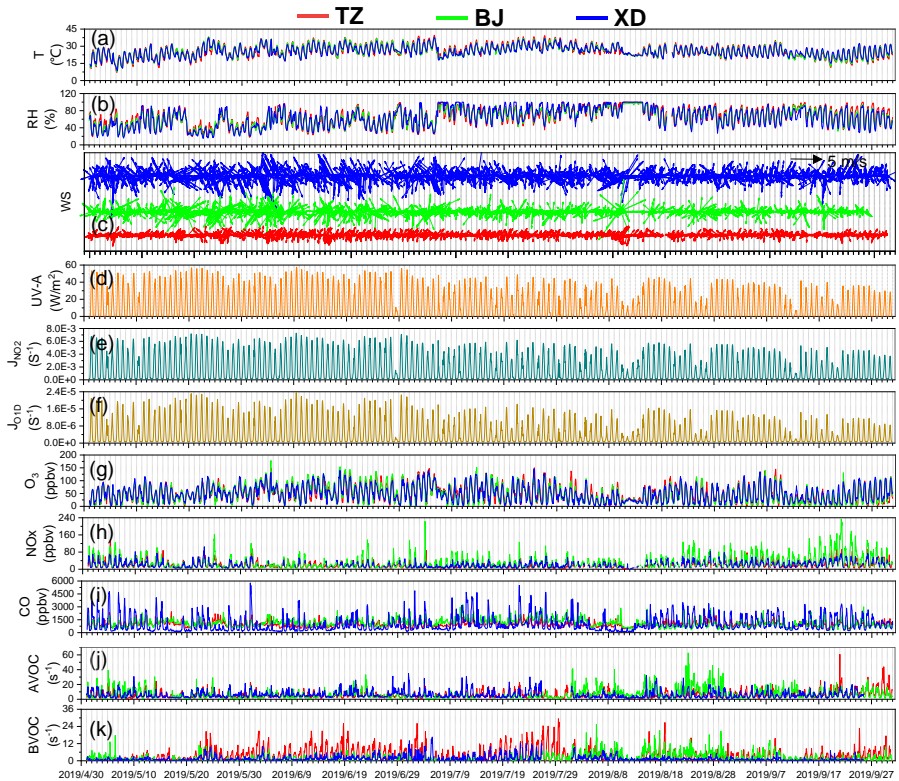

**Figure 2.** Time series of meteorological parameters, O₃ and its precursors (i.e., CO, NOx, VOCs)
throughout the whole campaign at the three sites in Zibo.






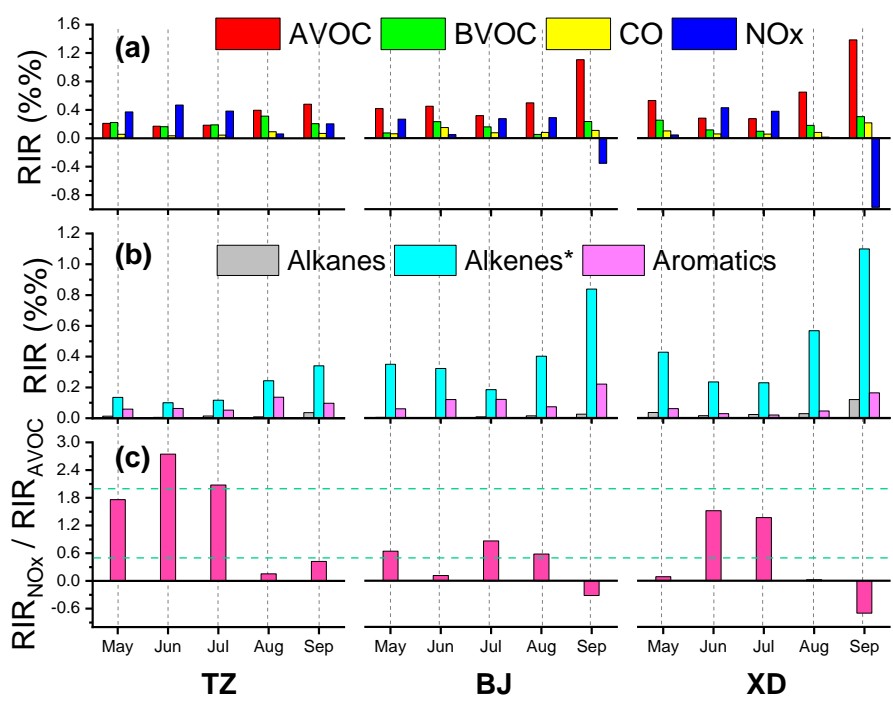


**Figure 3.** Time series of month-to-month RIR values of major precursor groups and $RIR_{NOx}/RIR_{AVOC}$ at
three sites (TZ, BJ and XD) in Zibo. The green dash line indicates to $RIR_{NOx}/RIR_{AVOC}$ = 0.5 and 2.






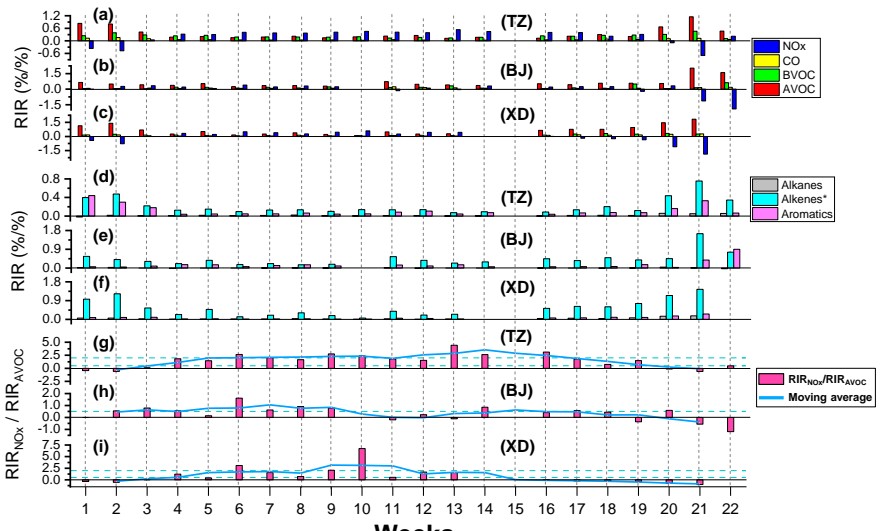

**Figure 4.** Time series of week-to-week RIR values of major precursor groups and $RIR_{NOx}/RIR_{AVOC}$ at
three sites (TZ, BJ, and XD) in Zibo. The blue lines in (g)-(i) are the three points moving average of
$RIR_{NOx}/RIR_{AVOC}$ value.

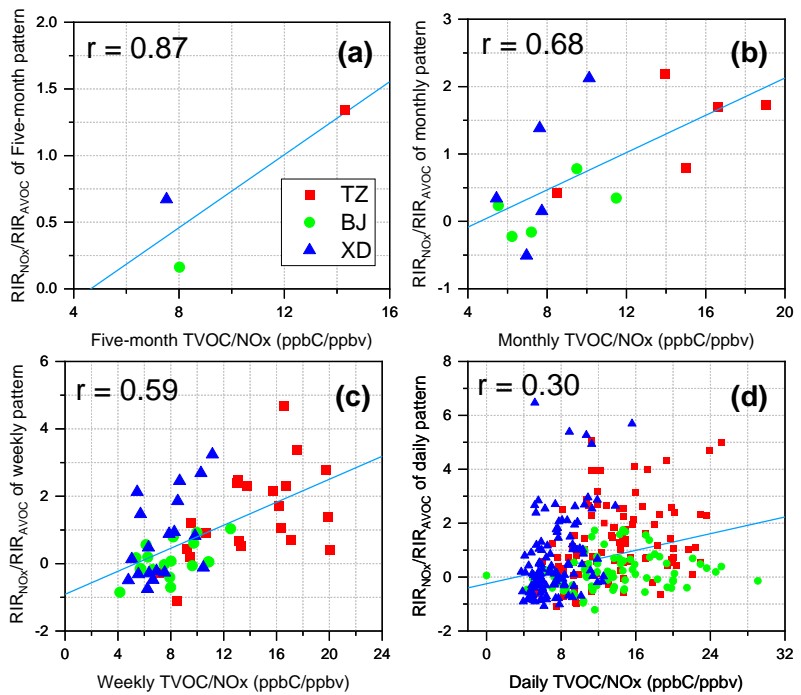

**Figure 5.** The correlations of TVOC/NOx with $RIR_{NOx}/RIR_{AVOC}$ at multiple patterns of time scale at the three sites in Zibo.






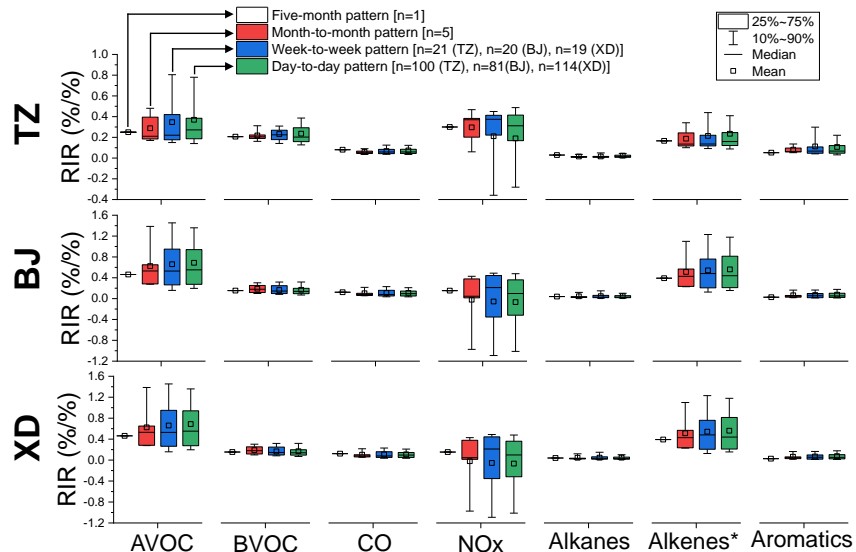

**Figure 6.** Distribution of RIR values of major precursor groups in multiple patterns of time scale at three
sites (TZ, BJ, and XD) in Zibo.




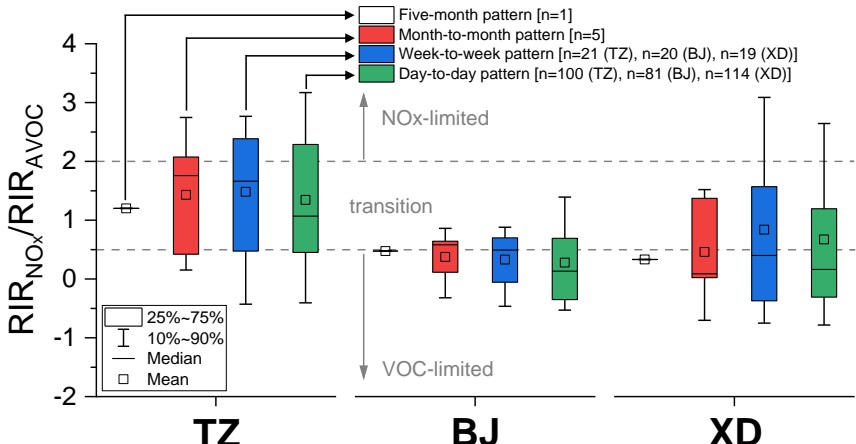


**Figure 7.** Distribution of $RIR_{NOx}/RIR_{AVOC}$ (indicator of photochemical regime) in multiple patterns of
time scale at three sites (TZ, BJ, and XD) in Zibo.



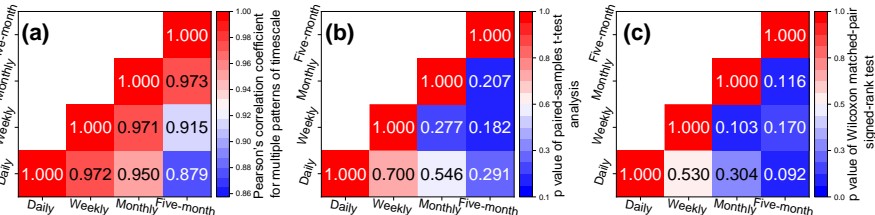

**Figure 8.** The statistical analysis results of RIR values (from Table S6) at multiple patterns of time scale: (a) Pearson's r correlation analysis (all the results have passed statistical significance assumed at $p <$ 0.01), (b) Paired-samples t-test analysis (*$p$ values refer to differences with a statistical significance assumes at $p <$0.05), (c) Wilcoxon matched-pair signed-rank test (*$p$ values refer to differences with a statistical significance assumes at $p <$0.05).





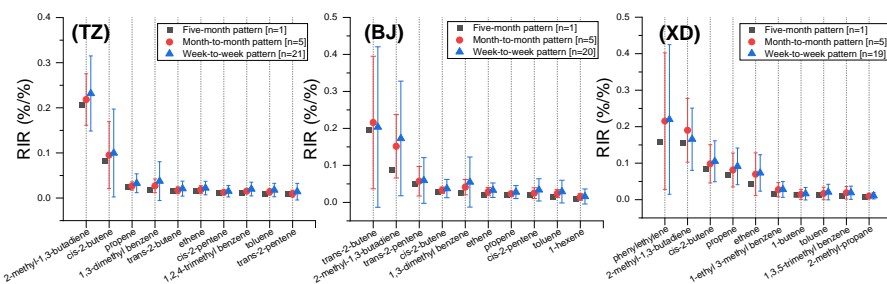

**Figure 9.** Averaged RIR values of individual AVOC species (top 10) at different patterns of time scale at three sites (TZ, BJ, and XD) in Zibo. The error bars represent the standard deviations of the mean.

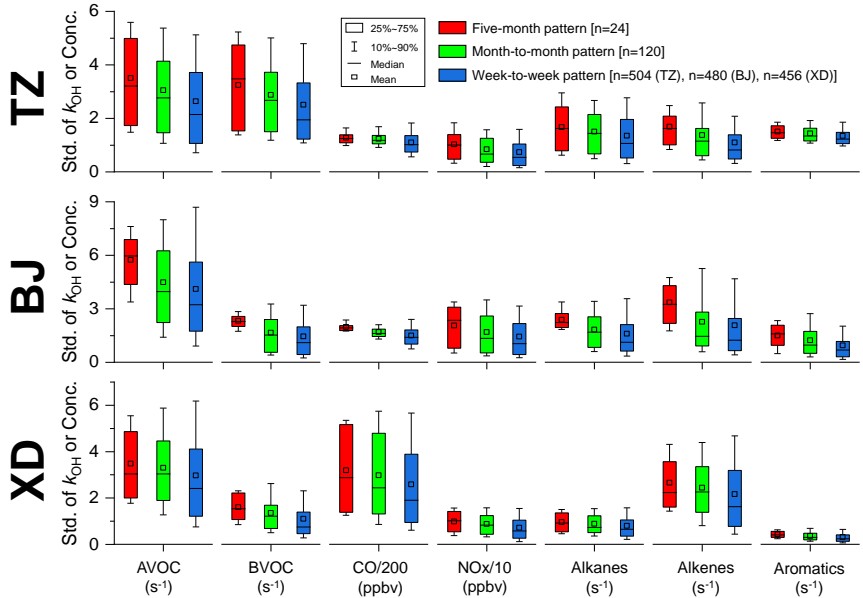

918

**Figure 10.** Distributions of the standard deviations (Std.) for OH reactivity ($k_{OH}$) or concentration of $O_3$

precursor groups for multiple patterns of time scale at the three sites in Zibo. For example, there would

be 24 standard deviation values when averaging into five-month diurnal patter; and months×24 standard

deviation values (n=120 for all sites) when averaging into monthly pattern; and weeks×24 standard

deviation values (n=504, 480, 456 for TZ, BJ, XD) when averaging into weekly pattern.