# Peer review of "O3-precursor relationship over multiple patterns of time scale"

_Atmospheric Chemistry and Physics, 2022_

## Referee Comment (RC2)

**O$_3$-precursor relationship over multiple patterns of time scale: A case study in Zibo, Shandong Province, China**

*Zhensen Zheng et al.*

This manuscript describes a recent 5-month field campaign to better understand the NOx-VOC sensitivity of ozone during the summer months in Zibo, China. The authors conducted 0-D box modeling with the MCM v3.3.1 near-explicit mechanism, and determined that the selected time scale for this modeling (i.e. daily, weekly, monthly, or campaign-wide) can affect the magnitude of the dependence of ozone formation on its precursors, with shorter time scales (i.e. daily) leading to a wider range of relative incremental reactivities (RIR). RIRs determine the expected effect of reducing certain emissions on the production of ozone, so are a useful tool in mitigating ozone pollution. The authors determine that the RIRs can vary within a city, due to local emissions, and can also vary within a season. This indicates that care should be taken when ascribing meaning to RIRs, as they can be dependent on the modeling parameters selected.

In general, I think this paper is worth publication in ACP. The main result, that averaging over several months blurs out some relevant chemical complexity, isn't that surprising, but the paper is well written, and it is good to see these results analyzed so completely and clearly. I think it will help the community choose time scales wisely.

I have a few minor suggestions/comments that should be addressed prior to publication:

**General comments:**

1) What is the authors recommendation for time scale? On line 35, they state that "integrating multiple patterns of time scale is useful to derive reliable and robust O3-precursor relationships". Do the authors not think that the daily time scale is the most accurate because of its detail? What would be the benefit of doing a weekly or monthly average instead?

2) A dilution parameter of 3/86400 s-1 was chosen based on its best fit to the data. Is this the only parameter that was tuned to fit the model to the data? Is it possible that the trends determined here could be affected by the selection of that dilution parameter?

3) In general, some of the Supplemental Figures and Tables are not presented in the order in which they appear in the manuscript.

4) On line 392, the authors state that the RIR_AVOC (and others) are increasing as the time scale gets narrower. But looking at Figure 6, it seems that only the mean is increasing. The median stays the same, implying that there are some more extreme values of RIR_AVOC in the daily model, that are getting averaged out as the time scale broadens. I think this is a different statement than saying that the RIR_AVOC value changes, and the authors should be more careful about making that distinction.

5) Section 3.7 describes the uncertainty analysis, but I believe the authors are conflating the mathematical terms for uncertainty (as analyzed by standard deviation of the averages), and the broader qualitative term for uncertainty (that some chemical species weren't included in the model). Figure 10 does not seem to be informative, it simply demonstrates that more data reduces the uncertainty, but that's well known. I would recommend removing that figure altogether, and focusing section 3.7 on what is missing in the model.

6) The RIR_CO is presented in many figures but never discussed. Does CO play a relevant role in the O3 formation here? If not, why is it presented in these figures?

**Specific comments:**

Line 19 and 91: "integrating" → "integrates"

Line 21: "multiple-site" → "multiple sites"

Line 26: The authors state that the RIRs are "consistent with time scale", but the manuscript demonstrates that they have different magnitudes. Do they mean consistent with regard to sign (i.e. all positive or all negative)? This should be stated more clearly. Same comment for line 33, which describes consistency in the photochemical regimes, when I think the authors mean consistency in the sign, but not the magnitude.

Line 58: Describing the 0-D model as "advanced" makes it sound like it is more complex than the regional scale air quality models on line 55. But I would say that the benefit of the 0-D model is its relative simplicity (despite the larger MCM mechanism), which allows the kind of in-depth chemical analysis that the authors do here.

Line 112: The Thermo Scientific 42i measures total NOx or NO. Can you describe more fully how the measured NOx was separated into NO and NO2? Do you expect any uncertainty there to make a difference to your modeling results?

Line 116: More information about the Zibo Eco-Environmental Monitoring Center is needed here. Is this data publicly available? What instrumentation was used? Or is there a reference to this site?

Line 125: "Tenax GR" → "Tenax GR cartridges"

Lines 128 – 137: It would be helpful to describe what differences might be expected between the FID and FID/PID instruments? Have they been cross-checked and calibrated?

Line 139: Were the 5 point calibration standards from 5 separate standard cylinders? Or dilution one cylinder. If the second option, does that affect the accuracy of the calibration at all?

Line 155: Define F0AM

Line 174: "due to significant miss" → "due to significant missing data". Additionally, which data was missing? It is hard to see in Figure 2 what is missing. Is the cause of the missing data related to the fact that these time periods seem to overlap to unusual (i.e. not very diurnal) patterns in the O3, such as Jun 3 – 6, Jul 5 – 8, and Aug 9 – 15 in figures S6 - 8?

Line 214: In this analysis, was the 10% change in X a 10% increase or a 10% decrease? Does it matter which is selected, since O3 has a non-linear response to NOx in particular?

Line 228: Do the authors have a hypothesis for why the wind speeds were so different at the different sites? Were they at equal altitudes above the ground? Could it have been an instrument issue?

Line 234: How was the scaling done?

Line 243: "quantity" → "quantify"

Line 253: See my general comment #2. Isn't model performance also due to the selection of the dilution parameter that maximizes model performance?

Line 287: Why is there no section dedicated to discussing the full campaign time scale? I only see three sections (monthly, weekly, and daily), but later on the authors discuss the four different time scale analyses.

Line 289 – 295: This is helpful information for understanding how the model was run, and should be moved to the methods section. This is also true for lines 314 – 319.

Line 396: How are these underestimation percentages being calculated?

Line 530: While it is true in this case that all time scales yielded the same information for ranking the top-10 VOC contributors, do the authors expect it would be true in all cases? Unless they have done that analysis, I would recommend changing the language here to just describe these results, and not try to make this broad statement about all models.

Line 546: "difference" → "differences"

Table 1: This is a summary of the most relevant 0-D box models, correct? In that case it should be stated as "Summary of relevant published 0-D box model studies", so it doesn't imply this is every box model ever published.

Text S1: This isn't ever reference in the main text, and doesn't add any new information. I would delete it.

Line S88: This URL should be a cited reference instead.

Table S2: State in the caption what the asterisk next to "alkenes" refers to. Additionally, does "non-listed in box model" mean that they were measured but not modeled? Or modeled but not measured?

---

## Author Comment (AC1)

**Response to reviewer #1's comments**

Journal: Atmos. Chem. Phys

Manuscript ID: acp-2022-592

Title: " O3-precursor relationship over multiple patterns of time scale: A case study in Zibo, Shandong Province, China"

Author(s): Zheng et al.

**General comments:**

Based on 5-month observation data of VOCs, CO, $NO_x$ and meteorological factors at three sites in a major prefecture-level city of Zibo, Shandong province of China, this manuscript explored the relationship between $O_3$ and its two precursors (VOCs and $NO_x$) by using a 0-D box model. The results implied that diagnosis of photochemical $O_3$ formation regimes was better based on model simulations with constrain of the observation data on shorter time scales (e.g., daily or weekly scales), which would have a certain significance for developing $O_3$ control strategies in different pollution areas. To my knowledge, there are few studies to comparably investigate the difference of photochemical $O_3$ formation regimes diagnosed by model simulations with input of observation data treated on different time scales. Therefore, this reviewer recommends the manuscript to be published in the journal after considering the following specifics.

**Reply:** We appreciate the professional and positive comments by the reviewer, and we have addressed the proposed concerns in below point-by-point, with revised text in red.

**Specifics:**

There are many grammar mistakes, repetitive specifications and unclear descriptions thorough the whole manuscript, and thus an English native speaker is suggested to polish the manuscript.

The results and discussion seemed to be very confused for discussing the results at the three sites on the four-time scales. As the topic of this manuscript is "$O_3$-precursor relationship over multiple patterns of time scale", the investigation is better focused on the observation data at one sampling site, rather than at the three sampling sites.

**Reply:** Thanks for the careful review of our manuscript. We have carefully checked and revised to improve our manuscript, and this manuscript has been polished by an English native speaker during the revision.

Moreover, we agree with the reviewer that the investigation may be better focused on the observation data at one sampling site to illustrate this topic of timescale issue. In this study, the three observation sites are co-located within Zibo city of Shandong province, and the distances among them are around 50 km. The field measurement was simultaneously carried out at the three sites in Zibo City during the whole campaign, and the analysis together with the three sites can provide more comprehensive results with synchronous and informative O3-NOx-VOC sensitivity. For example, the results from the three sites jointly showed that the discrepancy of O3-precursor relationship would become larger along with the time scale changed from narrower (i.e., daily scale) to wider (i.e., five-month scale) pattern, which reinforces the fact that using narrower time scale to derive the O3-precursor relationship would be more reliable and robust.

**Abstract:**

**Comment 1**: Line 24-27, the subject of "reactivity" doesn't match the predicate of "were" in this sentence; "time scale" is better replaced by "time scales"; "varied from wider and narrower" is better moved before the parentheses. This sentence seemed to be vague, and thus is better to be rephrased.

**Reply:** This sentence has been rephrased in our manuscript as below.

**Line 27-31**: "It was found that the relative incremental reactivity (RIR) of major precursor groups (e.g., anthropogenic volatile organic compound (AVOC), $NO_x$) was overall consistent in the sign along with time scales changed from wider to narrower (four patterns: five-month, monthly, weekly, and daily) at each site, though the magnitudes of RIR varied at different sites."

**Comment 2**: Lines 28-30, "The time series of the photochemical regime" seemed not match "magnitude".

**Reply:** We have corrected the sentence as below.

**Line 31-34**: "The time series of the photochemical regime (using $RIR_{NOx}/RIR_{AVOC}$ as indicator) in weekly or daily patterns further showed a synchronous temporal trend

among the three sites, while the magnitude of $RIR_{NOx}/RIR_{AVOC}$ was site-to-site dependent."

**Introduction:**

**Comment 1**: Lines 52-54, "the non-linearity of ozone pollution and complex process involved in it" should be "the complex non-linear relationship between $O_3$ formation and its precursors (VOCs and $NO_x$)"; "challenges" doesn't match "lies".

**Reply:** Corrected as below.

**Line 56-59**: "Given the complex non-linear relationship between $O_3$ formation and its precursors (VOCs and $NO_x$), challenges in mitigating its severity lie primarily in comprehensively understanding of $O_3$-precursor relationship (Su et al., 2018a; Tan et al., 2018a)."

**Comment 2**: Lines 61-65, the large spatiotemporal variability of $O_3$-precursor relationship has been widely reported in literature, rather than a finding of your recent study. Therefore, this sentence is better to be rephrased.

**Reply:** We fully agree with the reviewer's comments, and this sentence have been rephrased as below.

**Line 65-69**: "Some previous studies (Li et al., 2021; Lu et al., 2010a; Sicard et al., 2020; Yu et al., 2020b) have reported a large variability of $O_3$-precursor relationship in spatiotemporal scales in many cities of China, which indicates great challenges in current $O_3$ pollution control (Wang et al., 2017a; Xue et al., 2014b)."

**Reference:**

Li, K., Wang, X., Li, L., Wang, J., Liu, Y., Cheng, X., Xu, B., Wang, X., Yan, P., Li, S., Geng, C., Yang, W., Azzi, M. and Bai, Z.: Large variability of $O_3$-precursor relationship during severe ozone polluted period in an industry-driven cluster city (Zibo) of North China Plain, J. Clean. Prod., 316, 128252, doi:https://doi.org/10.1016/j.jclepro.2021.128252, 2021.

Yu, D., Tan, Z., Lu, K., Ma, X., Li, X., Chen, S., Zhu, B., Lin, L., Li, Y., Qiu, P., Yang, X., Liu, Y., Wang, H., He, L., Huang, X. and Zhang, Y.: An explicit study of local ozone budget and NOx-VOCs sensitivity in Shenzhen China, Atmos. Environ., 224, 117304, doi:https://doi.org/10.1016/j.atmosenv.2020.117304, 2020a.

Lu, K., Zhang, Y., Su, H., Brauers, T., Chou, C. C., Hofzumahaus, A., Liu, S. C., Kita, K., Kondo, Y., Shao, M., Wahner, A., Wang, J., Wang, X. and Zhu, T.: Oxidant (O$_3$ + NO2) production processes and formation regimes in Beijing, J. Geophys. Res. Atmos., 115(7), 1–18, doi:10.1029/2009JD012714, 2010a.

Lyu, X. P., Chen, N., Guo, H., Zhang, W. H., Wang, N., Wang, Y. and Liu, M.: Ambient volatile organic compounds and their effect on ozone production in Wuhan, central China, Sci. Total Environ., 541, 200–209, doi:https://doi.org/10.1016/j.scitotenv.2015.09.093, 2016a.

Xue, L. K., Wang, T., Gao, J., Ding, A. J., Zhou, X. H., Blake, D. R., Wang, X. F., Saunders, S. M., Fan, S. J., Zuo, H. C., Zhang, Q. Z. and Wang, W. X.: Ground-level ozone in four Chinese cities: Precursors, regional transport and heterogeneous processes, Atmos. Chem. Phys., 14(23), 13175–13188, doi:10.5194/acp-14-13175-2014, 2014c.

Wang, T., Xue, L., Brimblecombe, P., Lam, Y. F., Li, L. and Zhang, L.: Ozone pollution in China: A review of concentrations, meteorological influences, chemical precursors, and effects, Sci. Total Environ., 575, 1582–1596, doi:10.1016/j.scitotenv.2016.10.081, 2017a.

**Methods:**

**Comment 1**: Lines 107-112; 119-127; 128-137, why did you describe the VOCs measurements in three paragraphs? The time resolution of VOCs measurements was repeated for three times. The description of "a FID detector is applied for quantification" is not correct. The FID detector can only detect the signal of target species, rather than quantification of the targets. Why did you respectively select Tenax GR to pre-concentrate C6-C12 VOCs and C2-C6 VOCs for the GC-FID and the GC/FID/PID? Could the Tenax GR effectively capture C2-C6 VOCs at room temperature? What's the role of PID for the GC/FID/PID? How about variations of the retention times of VOCs during the monthly calibration period?

**Reply:** Thank you for your good comments.

We have simplified the description of VOCs instruments into one paragraph and removed the repetitive contents, as shown in **Line 108-130**. We have removed the incorrect sentence "FID detector is applied for quantification" in this revision.

Note that we intentionally did not have the technical options for setting up the commercial GC systems (also known as PAMS system), and some technical features (e.g., selection of FID or FID/PID detector; Tenax GR for VOC capture) were mainly determined by their manufacturers. Nevertheless, we tried to address some technical details metioned by the reviewer based on our limited instrument knowledge. For example, Tenax GR is a composite of the TENAX-TA matrix whereby 23% graphitized carbon is used as an integral part of the material, which is widely applied as a column packing material for trapping VOCs from the air for the commercial VOC instruments. As shown in previous studies, the Tenax GR cartridges can capature most volatile compounds (Brown et al., 1996), such as C5-C8 hydrocarbons (Cao et al., 1993), C2-C9 aldehydes and C3-C9 ketones (Lomonaco et al., 2018). Also, the Tenax GR can effectively capture C2-C6 VOCs at room temperature, while C6-C12 VOCs were pre-concentrated by cooling trap (range form -10°C to 10°C). Similar to FID, PID is another detector for target VOC species. We performed a single-point calibration (i.e., 6 ppbv) every month, and the retention times of measured VOCs remained overall consistent during the whole campaign.

**Line 116-131**: "Two online GC systems (gas chromatography–flame ionisation detector, GC-FID, Thermo Scientific GC5900) were deployed at TZ and BJ respectively to measure VOC species. For $C_2$-$C_5$ VOCs, desorption and separation were performed using a GC with pre-concentration on a combination of two columns, followed by a FID detector. For $C_6$-$C_{12}$ VOCs, air sample was pre-concentrated on Tenax GR cartridges and subsequently separated by chromatographic column, then detected by another FID detector. Similarly, one online system (gas chromatography–flame ionisation detector/photoionisation detector, GC-FID/PID, Syntech Spectras GC 955-615/815) was deployed at XD site. For $C_2$-$C_6$ VOCs, the hydrocarbons were concentrated on a Tenax GR carrier, then thermally desorbed and separated on a DB-1 column, and finally detected by FID and PID detectors. For $C_6$-$C_{12}$ VOCs, the air sample was concentrated on a Carbosieves SIII carrier at 5°C, then thermally desorbed and separated on a combination of two columns, and FID and PID detectors were employed for subsequent detection. These systems measured 55 VOC species at a 1-h resolution, and more detailed descriptions can be found elsewhere (Chien, 2007; Jiang et al., 2018; Xie et al., 2008)."

**Reference:**

Brown R H. What is the best sorbent for pumped sampling–thermal desorption of volatile organic compounds? Experience with the EC sorbents project[J]. Analyst, 1996, 121(9): 1171-1175.

Cao X L, Hewitt C N. Evaluation of Tenax-GR adsorbent for the passive sampling of volatile organic compounds at low concentrations[J]. Atmospheric Environment. Part A. General Topics, 1993, 27(12): 1865-1872.

Lomonaco T, Romani A, Ghimenti S, et al. Determination of carbonyl compounds in exhaled breath by on-sorbent derivatization coupled with thermal desorption and gas chromatography-tandem mass spectrometry[J]. Journal of Breath Research, 2018, 12(4): 046004.

Xie, X.; Shao, M.; Liu, Y.; Lu, S.; Chang, C.-C.; Chen, Z.-M. Estimate of Initial Isoprene Contribution to Ozone Formation Potential in Beijing, China. Atmos. Environ. 2008, 42 (24), 6000–6010.

Chien, Y.C. Variations in Amounts and Potential Sources of Volatile Organic Chemicals in New Cars. Sci. Total Environ. 2007, 382 (2), 228–239. https://doi.org/https://doi.org/10.1016/j.scitotenv.2007.04.022.

Jiang, M.; Lu, K.; Su, R.; Tan, Z.; Wang, H.; Li, L.; Fu, Q.; Zhai, C.; Tan, Q.; Yue, D. Ozone Formation and Key VOCs in Typical Chinese City Clusters. Chinese Sci. Bull. 2018, 63 (12), 1130–1141.

**Comment 2**: Lines 138-145, besides the calibration, field comparison for the VOCs measurements by using the two types of GCs at one of the three sites is most important for the QA/QC. Did you conduct the comparison?

**Reply:** We fully agree with the importance of inter-comparison for the VOCs measurements, which should be done by using the two types of GCs at one of the three sites. Unfortunately, we did not conduct such comparison in our campaign, as these VOC instruments were separately deployed for routine operation at three different sites, and it is very difficult to relocate and maintain them in one site due to practical reasons. Nevertheless, these commercial GC systems were regarded as standard VOC instruments, and were regularly checked and maintained during the whole campaign to ensure good QA/QC. Besides, these VOC instruments were regularly calibrated by standard gases with 55 VOC species from the same cylinder (Linde Co., Ltd, USA). Therefore, we assume that the VOC datasets obtained at the three sites are overall reliable for subsequent analysis in this study.

**Line 134-141**: "Unfortunately, we did not conduct the inter-comparison between the GC-FID and GC-FID/PID instruments at the same site due to practical reasons, as these VOC instruments were separately deployed at the three different sites for continuous routine operation. To ensure the quality assurance / quantity control (QA/QC) of online VOC measurement, two five-point calibrations (i.e., 2, 4, 6, 8, 10 ppbv, dilution from one cylinder) for standard gases with 55 VOC species (Linde Co., Ltd, USA) were carried out in May and August of 2019 at the three sites."

**Comment 3**: Lines 163-165, "for the best reproduction of $O_3$" at the end of this sentence.

**Reply:** Done. (Line 169)

**Comment 4**: Lines 177-179, the subject of "dataset" doesn't match the predicate of "were"; Considering the repetition for classifying the four patterns of time scale, this sentence is suggested to "Specifically, the entire campaign data classified as four patterns of time scale were modeled as base runs."

**Reply:** Corrected. (Line 181-182)

**Comment 4**: Lines 211-216, to avoid confusing between the species of X and its concentration, POx(X) and POx(X-ΔX) are suggested to be POx(CX) and POx(CX-ΔCX); either "(ΔX, 10% of X in this study in accordance with the previous studies" or "Therefore, ΔC(X)/C(X) was 10% in this study" can be deleted to avoid repetition.

**Reply:** We have corrected this as below.

**Line 215-221**:

$$RIR(X) = \frac{[PO_x(CX) - PO_x(CX - \Delta CX)]/PO_x(CX)}{\Delta CX/CX}$$

Here, X is a specific precursor (i.e., $NO_x$, CO or grouped / individual VOC species), CX is the measured concentration of precursor X, and ΔCX is the hypothetical concentration change (ΔCX/CX = 10% in this study in accordance with the previous studies (Lyu et al., 2016; Wang et al., 2018)). $PO_x(CX)$ represents the simulated $O_x$ production rate in a base run, whereas $PO_x(CX–\Delta CX)$ is the simulated $O_x$ production in a second run with a hypothetical concentration change of species X.

**Results and discussion:**

**Comment 1**: Line 245, "within" should be "among".

**Reply:** Corrected. (Line 261)

**Comment 2**: Lines 247-250, the bracket is suggested to be moved after alkene*.

**Reply:** Corrected.

**Comment 3**: Lines 254-257, why were the nocturnal $O_3$ concentrations significantly underestimated by the model simulations (e.g., Fig. S3)?

**Reply:** The nocturnal ground $O_3$ concentrations are mainly influenced by the physical process, such as aggravating vertical mixing and horizontal transport from ozone-rich plumes (He et al., 2022), but not produced from atmospheric chemical process at nighttime due to no photochemical activities. Unlike the 3-D air quality model, 0-D box model usually simplifies the representation of the physical processes (i.e., deposition and advection), and focuses on modelling chemical process (Lu et al., 2010; Xu et al., 2021). Therefore, due to the lack of representing $O_3$ sources from physical tranport while maintaining the nighttime chemical consumption of $O_3$ (e.g., $O_3$+NO titration reaction), uncertainty is unavoidable in simulating nocturnal $O_3$ concentrations by box modelling to some extent, which may explain the $O_3$ underestimation in our study.

**Line 273-278**: "However, on some days the modeling results underestimated or overestimated the $O_3$ concentrations, particularly the underestimation of nocturnal $O_3$ concentrations. Such discrepancies between the simulated and observed $O_3$ were likely due to limitations in explicit representations of atmospheric and transport processes (i.e., the horizontal and vertical transport process of ground ozone) by 0-D modeling approach (Lyu et al., 2019; Yu et al., 2020b)."

**Reference:**

He, C., Lu, X., Wang, H., Wang, H., Li, Y., He, G., He, Y., Wang, Y., Zhang, Y., Liu, Y., Fan, Q., and Fan, S.: Unexpected high frequency of nocturnal surface ozone enhancement events over China: Characteristics and mechanisms, Atmos. Chem. Phys. Discuss. [preprint], https://doi.org/10.5194/acp-2022-310, in review, 2022.

Lu, K., Zhang, Y., Su, H., Brauers, T., Chou, C. C., Hofzumahaus, A., Liu, S. C., Kita, K., Kondo, Y., Shao, M., Wahner, A., Wang, J., Wang, X. and Zhu, T.: Oxidant

($O_3$ + NO2) production processes and formation regimes in Beijing, J. Geophys. Res. Atmos., 115(7), 1–18, doi:10.1029/2009JD012714, 2010a.

Xu D, Yuan Z, Wang M, et al. Multi-factor reconciliation of discrepancies in ozone-precursor sensitivity retrieved from observation- and emission-based models. Environment International. 2022 Jan; 158:106952. DOI: 10.1016/j.envint.2021.106952.

**Comment 4**: Lines 280-287, why was the model performance for TZ better than for XD and BJ?

**Reply:** We applied an optimized dilution rate of $3/86400 \text{ s}^{-1}$ for all simulated days, which is conductive to ensuring the rationality and comparability of modeled results at the three sites. We infer that this optimized dilution rate for non-constraint species in our model configuration may result in better model performance in TZ than the other two sites.

**Line 297-300**: "In summary, TZ showed the best performance of the box model simulation, followed by XD and BJ, regardless of any statistical metrics or different patterns of time scale, which may be associated with the optimized dilution rate for non-constraint species in our model configuration."

**Comment 5**: Line 292, alkenes* have been noted before, and thus the brackets in here can be deleted.

**Reply:** Done.

**Comment 6**: Lines 309-313, the relationship between the monthly variations of the species and the RIR is better to be discussed, or readers cannot understand why you present them in here?

**Reply:** Thank you for the good comments. In our revision, we have added some descriptions to illustrate the monthly variations of the measurements from these species as below.

**Line 323-330**: "Significant monthly variations of $O_3$, $NO_x$, CO, VOC reactivity and TVOC/$NO_x$ ratios (in ppbC/ppbv, as a widely used simple metric to determine the photochemical regime) (National Research Council, 1991) were also observed from May to September (see **Figure S9** and **Table S3**) at the three sites. For example, the BVOC reactivity in TZ showed highest level among the three sites during the whole

campaign, and the AVOC reactivity in BJ showed more considerable variations in different months, which indicated spatial and temporal variations of local primary emission for $O_3$ precursors in Zibo city."

**Comment 7**: Lines 314-316, "two regimes (i.e., VOC-limited and $NO_x$-limited) or" can be deleted because the three regimes are prevailingly adopted.

**Reply:** Done.

**Comment 8**: Lines 326-328, the correlation between the monthly TVOC/$NO_x$ and $RIR_{NOx}/RIR_{AVOC}$ would become worse when only one sampling site was considered. Therefore, Fig. 5b could not well explain the considerable variation of monthly $O_3$ formation chemistry.

**Reply:** We agree with the reviewer's comments that the correlation between the monthly TVOC/$NO_x$ and $RIR_{NOx}/RIR_{AVOC}$ would vary when each site was considered individually. As shown in **Figure 5**, we added the correlation between TVOC/$NO_x$ and $RIR_{NOx}/RIR_{AVOC}$ for each individual site. It seems that the correlation between TVOC/$NO_x$ and $RIR_{NOx}/RIR_{AVOC}$ for each site was overall consistent with the result by merging all data points from the three sites. Hence, we suppose that the variations of $O_3$ formation chemistry can be elucidated by the variability of $O_3$ precursors at the three sites to some extent. This has been incorporated into our manuscript as below.

**Line 336-339**: "**Figure 5b** shows good consistency between monthly TVOC/$NO_x$ and $RIR_{NOx}/RIR_{AVOC}$, suggesting that the changes of local emissions for $O_3$ precursors may partially explain the considerable variation of $O_3$ formation chemistry in different months."

[Figure]

**Figure 5.** The correlations of TVOC/NO$_x$ with RIR$_{NOx}$/RIR$_{AVOC}$ at multiple patterns of time scale at the three sites in Zibo.

**Comment 9**: Lines 344-347, the correlation between the weekly TVOC/NO$_x$ and RIR$_{NOx}$/RIR$_{AVOC}$ at one sampling site (Fig. 5c) was also weak for explaining the weakly variation of O$_3$ formation chemistry. Additionally, the data point with TVOC/NO$_x$ of zero for BJ in Fig. 5d is wrong, should be removed.

**Reply:** Thanks for your detailed review, and pointed out our mistake. The data point with TVOC/NO$_x$ of zero for BJ in **Figure 5d** has been removed. In addition, the weekly correlation between TVOC/NO$_x$ and RIR$_{NOx}$/RIR$_{AVOC}$ at each site was slightly low but overall consistent with the result merged by three sites (see above **Figure 5** in **Comments 8**). This suggests that the weekly variation of O$_3$ formation chemistry can be partially explained by the variability of O$_3$ precursors. This has been incorporated into our manuscript as below.

**Line 355-358**: "Given the moderate correlation between weekly TVOC/NO$_x$ and RIR$_{AVOC}$/RIR$_{NOx}$ (**Figure 5c**), the temporal variations of RIR values and O$_3$ formation chemistry at the three sites may be partially elucidated by the emission changes of O$_3$ precursors."

**Comment 10**: Lines 361-363: "Additionally, the time series of daily $RIR_{NOx}$/$RIR_{AVOC}$ (**Figure S11**) first increased and then decreased during the entire campaign, which was also consistent with that of weekly scale.", the description seemed not well reflect the time series of daily $RIR_{NOx}$/$RIR_{AVOC}$ in Fig. S11 with irregular variations.

**Reply:** Thank you for careful review of our manuscript. This sentence has been rephrased in our manuscript as below.

**Line 372-374**: "Additionally, the time series of daily $RIR_{NOx}$/$RIR_{AVOC}$ (**Figure S13**) showed more irregular variations in temporal trends during the entire campaign, though such temporal trends were overall consistent with that of weekly scale in **Figure 4 g-i**."

**Comment 11**: Lines 399-402, the model simulations with inputting the average values for the five-month scale would greatly mask the large temporal variations of species especially for meteorological factors (such as sunlight and temperature), which is the key reason for the discrepancy of RIR values between five-month scale and daily scale. It is not proper to explain the discrepancy of RIR by the uncertainty.

**Comment 13**: Line 460, besides the analyzed uncertainties, the uncertainty due variation of meteorological factors for the long period scales may play a more important role in the $O_3$ sensitive chemistry.

**Reply** to **Comment 11** and **Comment 13:** Thank you for the good comments. We agree that both precursor emissions and meteorological factors play a key role in $O_3$ formation over a long observational period. Indeed, the averaged dataset for model input will mask the temporal variations of $O_3$ precursors and meteorological factors, and the exent of which depends on the selected timescale pattern. We rephrased some discussions in the revised manuscript as below.

**Line 478-480**: "This averaging approach will conceal the temporal variations of $O_3$ precursors and meteorological factors, particularly for a long-term observational campaign."

**Line 485-489**: "In addition, meteorological factors such as temperature and irradiation also play an important role on $O_3$ formation, especially these meteorological parameters can vary greatly over a long observational period (Boleti et al., 2020; Liu et

al., 2019b; Weng et al., 2022). Therefore, the masked temporal variation of these meteorological factors behind the averaged input dataset would also result in model uncertainty."

**Reference**:

Boleti, E., Hueglin, C., Grange, S. K., Prévôt, A. S. H., and Takahama, S.: Temporal and spatial analysis of ozone concentrations in Europe based on timescale decomposition and a multi-clustering approach, Atmos. Chem. Phys., 20, 9051–9066, https://doi.org/10.5194/acp-20-9051-2020, 2020.

Weng, X., Forster, G. L., and Nowack, P.: A machine learning approach to quantify meteorological drivers of ozone pollution in China from 2015 to 2019, Atmos. Chem. Phys., 22, 8385–8402, https://doi.org/10.5194/acp-22-8385-2022, 2022.

Liu, X., Lyu, X., Wang, Y., Jiang, F., and Guo, H.: Intercomparison of $O_3$ formation and radical chemistry in the past decade at a suburban site in Hong Kong, Atmos. Chem. Phys., 19, 5127–5145, https://doi.org/10.5194/acp-19-5127-2019, 2019.

**Comment 12**: Lines 415-418, there is a repeated comparison.

**Reply:** Thank you for careful review of our manuscript, and we have removed the repeated comparision.

---

## Author Comment (AC2)

**Response to reviewer #2's comments**

Journal: Atmos. Chem. Phys

Manuscript ID: acp-2022-592

Title: " $O_3$-precursor relationship over multiple patterns of time scale: A case study in Zibo, Shandong Province, China"

Author(s): Zheng et al.

**Overall comment**:

This manuscript describes a recent 5-month field campaign to better understand the $NO_x$-VOC sensitivity of ozone during the summer months in Zibo, China. The authors conducted 0-D box modeling with the MCM v3.3.1 near-explicit mechanism, and determined that the selected time scale for this modeling (i.e., daily, weekly, monthly, or campaign-wide) can affect the magnitude of the dependence of ozone formation on its precursors, with shorter time scales (i.e., daily) leading to a wider range of relative incremental reactivities (RIR). RIRs determine the expected effect of reducing certain emissions on the production of ozone, so are a useful tool in mitigating ozone pollution. The authors determine that the RIRs can vary within a city, due to local emissions, and can also vary within a season. This indicates that care should be taken when ascribing meaning to RIRs, as they can be dependent on the modeling parameters selected.

In general, I think this paper is worth publication in ACP. The main result, that averaging over several months blurs out some relevant chemical complexity, isn't that surprising, but the paper is well written, and it is good to see these results analyzed so completely and clearly. I think it will help the community choose time scales wisely.

**Reply:** We appreciate the professional and positive comments by the reviewer, and we have addressed the proposed concerns in below point-by-point, with revised text in red.

**General comments:**

**Comment 1**: What is the authors recommendation for time scale? On line 35, they state that "integrating multiple patterns of time scale is useful to derive reliable and robust $O_3$-precursor relationships". Do the authors not think that the daily time scale is

the most accurate because of its detail? What would be the benefit of doing a weekly or monthly average instead?

**Reply:** Thanks for pointing this unclear statement. Of course, we agree that the daily time scale is the most accurate because it can provide more informative details among the four patterns of time scale. We recommend the narrower time scale (i.e., daily pattern in this study) in box modelling, as it can provide a more reliable and robust O3-precursor relationship, when considering the non-negligible variability among the four patterns of time scale. We have revised this sentence as below.

**Line 40-41**: "This implies that utilizing narrower time scale (i.e., daily pattern) is useful to derive reliable and robust $O_3$-precursor relationship."

**Comment 2**: A dilution parameter of 3/86400 s$^{-1}$ was chosen based on its best fit to the data. Is this the only parameter that was tuned to fit the model to the data? Is it possible that the trends determined here could be affected by the selection of that dilution parameter?

**Reply:** In this study, the dilution rate is indeed the only parameter that was tuned to fit the model to the measured $O_3$ data. As described in **Text S1**, we have performed a stepwise sensitivity test to generate an optimal dilution rate for all non-constraint species for all simulation days. Many 0-D box model simulations include this dilution rate for all non-constraint species to avoid secondary species from building up to unreasonable levels (Bloss, et al., 2005; Wolf, et al., 2012; Wolf, et al., 2016), which is regarded as a technical model parameterization because of the very rare ideal stagnant conditions in the realistic atmsphere. As proposed in Wolf et al., (2016), the dilution rate ($k_{dil}$) is represented as a first order reaction in the box model. Currently we are unable to analyze how different $k_{dil}$ values affect the trends in photochemical regime as it requires systematical model calculation effort, but this is worth for further investigation in the future. These have been incorporated into our manuscript as below.

**Line S86-S93** in **Supplement Text S1**: "By comparing the modeled $O_3$ with observed $O_3$ for the three sites, we obtained an optimized $k_{dil}$ of 3/86400 s$^{-1}$, and assigned it to all non-constraint species for all simulation days, which is the only model parameter that was tuned to fit the measured $O_3$ data. In general, this optimized $k_{dil}$ is conductive to ensuring the rationality and comparability of model performance for all

modeled days at the three sites, and it is also worth for further investigation about how different $k_{dil}$ values affect the trends in photochemical regime."

**Reference:**

Bloss, C.; Wagner, V.; Jenkin, M. E.; Volkamer, R.; Bloss, W. J.; Lee, J. D.; Heard, D. E.; Wirtz, K.; Martin-Reviejo, M.; Rea, G.; Wenger, J. C.; Pilling, M. J. Development of a Detailed Chemical Mechanism (MCMv3.1) for the Atmospheric Oxidation of Aromatic Hydrocarbons. *Atmos. Chem. Phys.* **2005**, *5* (3), 641–664. https://doi.org/10.5194/acp-5-641-2005.

Jenkin, M. E.; Wyche, K. P.; Evans, C. J.; Carr, T.; Monks, P. S.; Alfarra, M. R.; Barley, M. H.; McFiggans, G. B.; Young, J. C.; Rickard, A. R. Development and Chamber Evaluation of the MCM v3.2 Degradation Scheme for β-Caryophyllene. *Atmos. Chem. Phys.* **2012**, *12* (11), 5275–5308. https://doi.org/10.5194/acp-12-5275-2012.

Wolfe, G. M.; Marvin, M. R.; Roberts, S. J.; Travis, K. R.; Liao, J. The Framework for 0-D Atmospheric Modeling (F0AM) v3. 1. *Geosci. Model Dev.* **2016**, *9* (9), 3309–3319.

**Comment 3**: In general, some of the Supplemental Figures and Tables are not presented in the order in which they appear in the manuscript.

**Reply:** Thanks for your careful review of our manuscript, and we have checked and relocated these supplemental Figures and Tables in order.

**Comment 4**: On line 392, the authors state that the RIR_AVOC (and others) are increasing as the time scale gets narrower. But looking at Figure 6, it seems that only the mean is increasing. The median stays the same, implying that there are some more extreme values of RIR_AVOC in the daily model, that are getting averaged out as the time scale broadens. I think this is a different statement than saying that the RIR_AVOC value changes, and the authors should be more careful about making that distinction.

**Reply:** Thanks for the good comments. As mentioned by the reviewer in **Figure 6**, there are discrepancies of the RIRs trends depicted by mean and median respectively. To make it clear, we use the mean to describe the trends of RIRs in accordance with the method of averaging dataset into multiple time scales throughout this section. This has been incorporated into our manuscript as below.

**Line 403-410**: "As the time scale changed from wider (i.e., five-month scale) to narrower (i.e., daily scale) pattern, all three sites showed increases in the means of $RIR_{AVOC}$ and $RIR_{alkenes*}$ as well as decreases in averaged $RIR_{NOx}$, whereas the averaged RIR of other precursors (i.e., BVOC, CO, alkanes and aromatics) did not vary obviously (see **Table S6**). Comparing with the $O_3$-VOC-$NO_x$ sensitivity at the daily scale, the results obtained at the five-month scale underestimated $O_3$-AVOC sensitivity (indicated by averaged RIR values) by 48% (TZ), 66% (BJ), and 49% (XD), and overestimated $O_3$-$NO_x$ sensitivity by 37% (TZ), 142% (BJ), and 144% (XD)."

**Line 428-431**: "Compared with the five-month pattern, it was further found that the averaged $RIR_{NOx}/RIR_{AVOC}$ from other time scale patterns (i.e., monthly, weekly, and daily) were higher (12% to 20% for TZ; 38% to 153% for XD) or lower (21% to 65% for BJ) than that from five-month scale."

**Comment 5**: Section 3.7 describes the uncertainty analysis, but I believe the authors are conflating the mathematical terms for uncertainty (as analyzed by standard deviation of the averages), and the broader qualitative term for uncertainty (that some chemical species weren't included in the model). Figure 10 does not seem to be informative, it simply demonstrates that more data reduces the uncertainty, but that's well known. I would recommend removing that figure altogether, and focusing section 3.7 on what is missing in the model.

**Reply:** We agree with the reviewer, and we have removed the **Figure 10** in manuscript and relocated into supplement as Figure S14. To focus the uncertainty on what is missing in the model, we have simplified and rephrased the uncertainty analysis about averaging approach in **Section 3.7** of our manuscript as below.

**Line 480-485**: "**Figure S14** shows the distributions of the standard deviations for OH reactivity ($k_{OH}$) or concentration of $O_3$ precursor groups at three averaged patterns of time scale at the three sites. As the time scale changed from wider (i.e., five-month scale) to narrower (i.e., weekly scale) pattern, the uncertainty (indicated by the average, median and 25%-75% quantile) decreased accordingly."

**Comment 6**: The RIR_CO is presented in many figures but never discussed. Does CO play a relevant role in the $O_3$ formation here? If not, why is it presented in these figures?

**Reply:** Overall, CO played a relatively limited role in comparison with other major categories of $O_3$ precursors (e.g., $NO_x$, AVOC and BVOC) at the three sites. Therefore, those RIR_CO figures without discussion in supplement have been removed directly, while we have added some descriptions for RIR_CO plots in manuscript (see below).

**Line 316-318**: "In addition, the $RIR_{CO}$ values at the three sites suggested its limited role in $O_3$ formation at the three sites, compared with other major categories of $O_3$ precursors."

**Line 403-407**: "As the time scale changed from wider (i.e., five-month scale) to narrower (i.e., daily scale) pattern, all three sites showed increases in the means of $RIR_{AVOC}$ and $RIR_{alkenes*}$ as well as decreases in averaged $RIR_{NOx}$, whereas the averaged RIR of other precursors (i.e., BVOC, CO, alkanes and aromatics) did not vary obviously (see **Table S6**)."

**Specific comments:**

**Comment 1**: Line 19 and 91: "integrating" → "integrates"

**Reply:** Corrected. (Line 22 and 95)

**Comment 2**: Line 21: "multiple-site" → "multiple sites"

**Reply:** Corrected. (Line 24)

**Comment 3**: Line 26: The authors state that the RIRs are "consistent with time scale", but the manuscript demonstrates that they have different magnitudes. Do they mean consistent with regard to sign (i.e., all positive or all negative)? This should be stated more clearly. Same comment for line 33, which describes consistency in the photochemical regimes, when I think the authors mean consistency in the sign, but not the magnitude.

**Reply:** Thanks for careful review of our manuscript, and we have rephrased the two sentences as below.

**Line 27-31**: "It was found that the relative incremental reactivity (RIR) of major precursor groups (e.g., anthropogenic volatile organic compound (AVOC), $NO_x$) was overall consistent in the sign along with time scales changed from wider to narrower

(four patterns: five-month, monthly, weekly, and daily) at each site, though the magnitudes of RIR varied at different sites."

**Line 36-39**: "It was further found that the campaign-averaging photochemical regimes showed overall consistency in the sign but non-negligible variability among the four patterns of time scale, which was mainly due to the embedded uncertainty in model input dataset when averaging individual daily pattern into different timescales."

**Comment 4**: Line 58: Describing the 0-D model as "advanced" makes it sound like it is more complex than the regional scale air quality models on line 55. But I would say that the benefit of the 0-D model is its relative simplicity (despite the larger MCM mechanism), which allows the kind of in-depth chemical analysis that the authors do here.

**Reply:** We agree with the reviewer's comments that the regional scale air quality models are more sophisticated than 0-D box model, and we have revised this statement in our manuscript as below.

**Line 62-65**: "Unlike the complicated 3-D air quality models, the 0-D box model is an observation-based model that implemented with gas-phase chemical mechanism, and has been widely used to diagnose $O_3$-precursor relationship in various locations (Liu et al., 2021a; Sun et al., 2016; Tan et al., 2019b; Xue et al., 2014a; Yu et al., 2020a)."

**Comment 5**: Line 112: The Thermo Scientific 42i measures total $NO_x$ or NO. Can you describe more fully how the measured $NO_x$ was separated into NO and NO2? Do you expect any uncertainty there to make a difference to your modeling results?

**Reply:** Thank you for the good comments. The overestimated $NO_2$ by chemiluminescence technique is always a challenging problem, therefore some studies also designated "$NO_2$" as "$NO_y$-NO". Both gaseous $HNO_3$ and organic nitrates can result in interferences on $NO_x$ measurement by chemiluminescence technique, and they are typically found in some polluted urban environments. For example, gaseous $HNO_3$ contributed approximately most to total nitrate (particle-phase and gas-phase nitrate) through gas-particle partitioning in summer (Ryota et al., 2022; Uno et al., 2017), while the contribution of organic nitrates to the total particle nitrate decreased as the $PM_{2.5}$ loading increased (Ge et al, 2022). In addition, the study of Xu et al. (2013) suggested that the overestimation of $NO_2$ by the molybdenum converter is limited in areas with fresh $NO_x$ emission sources, while such interference is more significant in rural or

remote areas due to large amount of oxidized nitrogen in aged air mass. Considering significant fresh $NO_x$ emissions nearby the three selected sites in Zibo, we believe that the interference of $NO_2$ measurement from chemiluminescence method should be limited in this study. Nevertheless, it is still meaningful to perform more in-depth study on $NO_x$ measurement uncertainty in box model simulation in the future, particularly when both traditional and accurate $NO_x$ measurement are available, as the accuracy of $NO_x$ measurement is essential in determing the photochemical regime. These discussions have been incorporated into our manuscript as below.

**Line 502-507**: "Besides, both gaseous $HNO_3$ and organic nitrates can result in interferences on $NO_x$ measurement by chemiluminescence technique, which may arise uncertainty in our box modelling (Ge et al., 2022; Uno et al., 2017; Xu et al., 2013). Since accurate $NO_x$ measurement is essential in determining the photochemical regime, more in-depth studies on $NO_x$ measurement uncertainty in box model simulation are required in the future."

**Reference**:

Ge D, Nie W, Sun P, et al. Characterization of particulate organic nitrates in the Yangtze River Delta, East China, using the time-of-flight aerosol chemical speciation monitor[J]. Atmospheric Environment, 2022, 272: 118927.

Steinbacher M, Zellweger C, Schwarzenbach B, et al. Nitrogen oxide measurements at rural sites in Switzerland: Bias of conventional measurement techniques[J]. Journal of Geophysical Research: Atmospheres, 2007, 112(D11).

Uno I, Osada K, Yumimoto K, et al. Seasonal variation of fine-and coarse-mode nitrates and related aerosols over East Asia: synergetic observations and chemical transport model analysis[J]. Atmospheric Chemistry and Physics, 2017, 17(23): 14181-14197.

Xu Z, Wang T, Xue L K, et al. Evaluating the uncertainties of thermal catalytic conversion in measuring atmospheric nitrogen dioxide at four differently polluted sites in China[J]. Atmospheric environment, 2013, 76: 221-226.

Nojiri R, Osada K, Kurosaki Y, et al. Variations in gaseous nitric acid concentrations at Tottori, Japan: Long-range transport from the Asian continent and local production[J]. Atmospheric Environment, 2022, 274: 118988.

**Comment 6**: Line 116: More information about the Zibo Eco-Environmental Monitoring Center is needed here. Is this data publicly available? What instrumentation was used? Or is there a reference to this site?

**Reply:** Sorry for the confusing statement, and Zibo Eco-Environmental Monitoring Center and our group are responsible for the routine operation of these monitoring sites in Zibo city. In this study, the meteorological parameters (i.e., temperature, relative humidity, UV-A solar radiation, precipitation, wind speed, and wind direction) were monitored by standard instruments following the Chinese meteorological monitoring regulation (GB/T 35221-2017). The dataset of the three sites in this study can be available after its publication, and we have added a relevant reference to these sites (Li et al., 2021).

**Line 113-116**: "Following the Chinese meteorological monitoring regulation (GB/T 35221-2017), we continuously monitored the meteorological parameters (i.e., temperature, relative humidity, UV-A solar radiation, precipitation, wind speed, and wind direction) at the three sites (Li et al., 2021)."

**Reference**:

Li, K., Wang, X., Li, L., Wang, J., Liu, Y., Cheng, X., Xu, B., Wang, X., Yan, P., Li, S., Geng, C., Yang, W., Azzi, M. and Bai, Z.: Large variability of $O_3$-precursor relationship during severe ozone polluted period in an industry-driven cluster city (Zibo) of North China Plain, J. Clean. Prod., 316, 128252, doi:https://doi.org/10.1016/j.jclepro.2021.128252, 2021.

**Comment 7**: Line 125: "Tenax GR" → "Tenax GR cartridges"

**Reply:** Corrected. (Line 121)

**Comment 8**: Lines 128-137: It would be helpful to describe what differences might be expected between the FID and FID/PID instruments? Have they been cross-checked and calibrated?

**Reply:** Thank you for the good comments, and another reviewer also pointed out similar issue. In **Methods Section**, we have briefly discussed the differences between the GC-FID and GC-FID/PID instruments. Unfortunately, we did not conduct the inter-comparison between the GC-FID and GC-FID/PID instruments at the same site, as

these VOC instruments were separately deployed at the three different sites for continuous routine operation, which is very difficult to relocate and maintain them in one site due to practical reasons. Nevertheless, these VOC instruments at the three sites are commercial instruments, and were regularly maintained and calibrated by standard gases with 55 VOC species from the same cylinder (Linde Co., Ltd, USA). Therefore, we assume the VOC datasets at the three sites are overall reliable and suitable for subsequent analysis in this study. These discussions have been incorporated into our manuscript as below.

**Line 116-131**: "Two online GC systems (gas chromatography–flame ionisation detector, GC-FID, Thermo Scientific GC5900) were deployed at TZ and BJ respectively to measure VOC species. For $C_2$-$C_5$ VOCs, desorption and separation were performed using a GC with pre-concentration on a combination of two columns, followed by a FID detector. For $C_6$-$C_{12}$ VOCs, air sample was pre-concentrated on Tenax GR cartridges and subsequently separated by chromatographic column, then detected by another FID detector. Similarly, one online system (gas chromatography–flame ionisation detector/photoionisation detector, GC-FID/PID, Syntech Spectras GC 955-615/815) was deployed at XD site. For $C_2$-$C_6$ VOCs, the hydrocarbons were concentrated on a Tenax GR carrier, then thermally desorbed and separated on a DB-1 column, and finally detected by FID and PID detectors. For $C_6$-$C_{12}$ VOCs, the air sample was concentrated on a Carbosieves SIII carrier at 5℃, then thermally desorbed and separated on a combination of two columns, and FID and PID detectors were employed for subsequent detection.   These systems measured 55 VOC species at a 1-h resolution, and more detailed descriptions can be found elsewhere (Chien, 2007; Jiang et al., 2018; Xie et al., 2008)."

**Line 134-137**: "Unfortunately, we did not conduct the inter-comparison between the GC-FID and GC-FID/PID instruments at the same site due to practical reasons, as these VOC instruments were separately deployed at the three different sites for continuous routine operation."

**Comment 9**: Line 139: Were the 5-point calibration standards from 5 separate standard cylinders? Or dilution one cylinder. If the second option, does that affect the accuracy of the calibration at all?

**Reply:** In our campaign, we selected the later (i.e., dilution from one cylinder) to perform the 5-point calibration in our campaign, and we have added this information into our manuscript as below. According to **Table S2**, the correlation coefficient of five-point calibration (i.e., 2, 4, 6, 8, 10 ppbv) for the 55 VOC species were nearly 0.9990, thus we assume the impact on accuracy of the calibration by the second option is relatively limited.

**Line 137-141**: "To ensure the quality assurance / quantity control (QA/QC) of online VOC measurement, two five-point calibrations (i.e., 2, 4, 6, 8, 10 ppbv, dilution from one cylinder) for standard gases with 55 VOC species (Linde Co., Ltd, USA) were carried out in May and August of 2019 at the three sites."

**Comment 10**: Line 155: Define F0AM.

**Reply:** Done. (Line 155-156)

**Comment 11**: Line 174: "due to significant miss" → "due to significant missing data". Additionally, which data was missing? It is hard to see in Figure 2 what is missing. Is the cause of the missing data related to the fact that these time periods seem to overlap to unusual (i.e., not very diurnal) patterns in the $O_3$, such as Jun 3-6, Jul 5-8, and Aug 9-15 in figures S6-8?

**Reply:** We have corrected the above statement as "due to some missing data". Indeed, we cannot perform simulation for some individual days due to some missing data as it requires a complete 24 h dataset for model input, which leads to limited modelling days [n=100 (TZ), n=81 (BJ), n=114 (XD)] as shown in **Figures S6-S8** for daily scale pattern.

**Comment 12**: Line 214: In this analysis, was the 10% change in X a 10% increase or a 10% decrease? Does it matter which is selected, since $O_3$ has a non-linear response to $NO_x$ in particular?

**Reply:** It is assumed as 10% decrease in X for RIR calculation. We believe that it does not matter either 10% increase or 10% decrease, as the calcuated RIR reflects a relative change of $O_3$ production rate to the change in X. In addition, we further assess the influence of choosing different hypothetical changes (i.e., 5%, 10%, and 15%) on RIR values, thus box model sensitivity test was performed with the above three

scenarios. **Figure 1** (in below) shows the model-derived RIR values under three hypothetical changes, using averaged diurnal pattern of five-month time scale as model input. In general, the RIR values of O₃ precursor groups and $RIR_{NOx}/RIR_{AVOC}$ ratios were overall consistent under different hypothetical changes. Since "10% decrease" was widely employed in the previous studies (Lyu et al., 2016; Wang et al., 2017a, 2018b), we applied 10% as hypothetical change in our RIR calculation for consistency.

[Figure]

Figure 1. The RIR values of O₃ precursor groups and $RIR_{NOx}/RIR_{AVOC}$ at different hypothetical changes (i.e., 5%, 10%, and 15%) using diurnal average of five-month pattern as model input at the three sites.

**Reference**:

Lyu, X. P., Chen, N., Guo, H., Zhang, W. H., Wang, N., Wang, Y. and Liu, M.: Ambient volatile organic compounds and their effect on ozone production in Wuhan, central China, Sci. Total Environ., 541, 200–209, doi:10.1016/j.scitotenv.2015.09.093, 2016.

Wang, Y., Wang, H., Guo, H., Lyu, X., Cheng, H., Ling, Z., Louie, P. K. K., Simpson, I. J., Meinardi, S. and Blake, D. R.: Long-term O₃-precursor relationships in Hong Kong: Field observation and model simulation, Atmos. Chem. Phys., 17(18), 10919–10935, doi:10.5194/acp-17-10919-2017, 2017.

Wang, Y., Guo, H., Zou, S., Lyu, X., Ling, Z., Cheng, H. and Zeren, Y.: Surface O₃ photochemistry over the South China Sea: Application of a near-explicit chemical mechanism box model, Environ. Pollut., 234, 155–166, doi:10.1016/j.envpol.2017.11.001, 2018.

**Comment 13**: Line 228: Do the authors have a hypothesis for why the wind speeds were so different at the different sites? Were they at equal altitudes above the ground? Could it have been an instrument issue?

**Reply:** The wind speeds at the three sites were measured following Chinese meteorological monitoring regulation (GB/T 35221-2017), thus we assume the obtained dataset of wind speeds should be reliable and highly unlikely due to instrument error. In addition, the altitudes above the ground of the three sites are nearly same. We guess such site-to-site difference of wind speeds may be associated with the discrepancies from local meteorological field, given the relatively long distance (i.e., more than 50 km) among the three sites.

**Comment 14**: Line 234: How was the scaling done?

**Reply:** Specifically, the geographical coordinates, date and time were initialized into the TUV model to derive photolysis rates and solar radiation. We obtained the scaling factor by comparing the observed with modeled solar radiation, and used this scaling factor to scale the TUV model-derived photolysis rates (Lyu et al., 2019; Lyu et al., 2016).

**Line 163-167**: "Specifically, the geographical coordinates, date and time were initialized into the TUV model to derive photolysis rates and solar radiation. We obtained the scaling factor by comparing the observed with modeled solar radiation, and used this scaling factor to scale the TUV model derived photolysis rates."

**Reference**:

Lyu, X., Wang, N., Guo, H., Xue, L., Jiang, F., Zeren, Y., Cheng, H., Cai, Z., Han, L. and Zhou, Y.: Causes of a continuous summertime $O_3$ pollution event in Jinan, a central city in the North China Plain, Atmos. Chem. Phys., 19(5), 3025–3042, doi:10.5194/acp-19-3025-2019, 2019.

Lyu, X. P., Chen, N., Guo, H., Zhang, W. H., Wang, N., Wang, Y. and Liu, M.: Ambient volatile organic compounds and their effect on ozone production in Wuhan, central China, Sci. Total Environ., 541, 200–209, doi:https://doi.org/10.1016/j.scitotenv.2015.09.093, 2016.

**Comment 15**: Line 243: "quantity" $\rightarrow$ "quantify"

**Reply:** Corrected. (Line 259)

**Comment 16**: Line 253: See my general comment #2. Isn't model performance also due to the selection of the dilution parameter that maximizes model performance?

**Reply:** Indeed, dilution rate is the only parameter that was tuned to maximize box model performance. We obtained an optimized dilution rate of $3/86400$ s$^{-1}$ through sensitivity test, and assigned it to all non-constraint species for all simulation days and the three sites (see details in **Text S1** and **reply to Commet 2**).

**Comment 17**: Line 287: Why is there no section dedicated to discussing the full campaign time scale? I only see three sections (monthly, weekly, and daily), but later on the authors discuss the four different time scale analyses.

**Reply:** The RIR results from five-month (full campaign) pattern of time scale have been comprehensively discussed in the **Section 3.6** by comparing it with other patterns of timescale.

**Comment 18**: Line 289 – 295: This is helpful information for understanding how the model was run, and should be moved to the methods section. This is also true for lines 314-319.

**Reply:** We agree with the reviewer's comments, and these two parts have been relocated in the **Methods Section** in our manuscript as below.

**Line 224-232**: "In this study, the $O_3$ precursors were divided into four major categories, including anthropogenic VOC (AVOC), biogenic VOC (BVOC, only isoprene in this study), CO and $NO_x$ (Tan et al., 2019b). AVOC was further divided into three subcategories: alkanes, aromatics and alkenes* (the asterisk denotes anthropogenic alkenes, excluding isoprene in this study) (Yu et al., 2020a). As mentioned, RIR method was applied mainly to evaluate the $O_3$-$NO_x$-VOC sensitivity and determine the photochemical regimes among four patterns of time scale. Thus, we calculated the RIR values of major precursor groups (i.e., AVOC, BVOC, CO, $NO_x$, alkanes, alkenes* and aromatics) to further quantify the $O_3$-precursor relationship."

**Line 233-238**: "In general, $O_3$ formation chemistry is usually classified into three regimes (i.e., VOC-limited, transitional and $NO_x$-limited) (He et al., 2019; Wang et al., 2018). In this study, $RIR_{NOx}/RIR_{AVOC}$ (the ratio of two RIR values) was used as a metric to classify the photochemical regimes (Li et al., 2021). Specifically, $RIR_{NOx}/RIR_{AVOC}$ value of less than 0.5 was defined as VOC-limited regime, greater than 2 as $NO_x$-limited regime, and from 0.5 to 2 as transitional regime (see **Text S2** and **Table S4**) (Li et al., 2021)."

**Comment 19**: Line 396: How are these underestimation percentages being calculated?

**Reply:** These underestimation percentages were calculated by comparing the averaged RIR from five-month scale with the mean of RIRs from daily patterns, and all data were summarized in **Table S6**. This has been incorporated into revised manuscript in below.

**Line 403-410**: "As the time scale changed from wider (i.e., five-month scale) to narrower (i.e., daily scale) pattern, all three sites showed increases in the means of $RIR_{AVOC}$ and $RIR_{alkenes*}$ as well as decreases in averaged $RIR_{NOx}$, whereas the averaged RIR of other precursors (i.e., BVOC, CO, alkanes and aromatics) did not vary obviously (see **Table S6**). Comparing with the $O_3$-VOC-$NO_x$ sensitivity at the daily scale, the results obtained at the five-month scale underestimated $O_3$-AVOC sensitivity (indicated by averaged RIR values) by 48% (TZ), 66% (BJ), and 49% (XD), and overestimated $O_3$-$NO_x$ sensitivity by 37% (TZ), 142% (BJ), and 144% (XD)."

**Line 428-430**: "Compared with the five-month pattern, it was further found that the averaged $RIR_{NOx}/RIR_{AVOC}$ from other time scale patterns (i.e., monthly, weekly, and daily) were higher (12% to 20% for TZ; 38% to 153% for XD) or lower (21% to 65% for BJ) than that from five-month scale."

**Comment 20**: Line 530: While it is true in this case that all time scales yielded the same information for ranking the top-10 VOC contributors, do the authors expect it would be true in all cases? Unless they have done that analysis, I would recommend changing the language here to just describe these results, and not try to make this broad statement about all models.

**Reply:** We agree with the reviewer's comments, and have removed this broad statement "This demonstrates that datasets with wider pattern of time scale can still produce an accurate RIR ranking / prioritization for VOC control" in this revision.

**Comment 21**: Line 546: "difference" → "differences"

**Reply:** Corrected. (Line 553)

**Comment 22**: Table 1: This is a summary of the most relevant 0-D box models, correct? In that case it should be stated as "Summary of relevant published 0-D box model studies", so it doesn't imply this is every box model ever published.

**Reply: Table 1** is indeed a summary of the most relevant 0-D box models, and we have corrected the Table 1 caption as "Summary of relevant published 0-D box model studies".

**Comment 23**: Text S1: This isn't ever reference in the main text, and doesn't add any new information. I would delete it.

**Reply:** We have deleted this part in Supplement.

**Comment 24**: Line S88: This URL should be a cited reference instead.

**Reply:** Done. (Line 100 in Supplement)

**Comment 25**: Table S2: State in the caption what the asterisk next to "alkenes" refers to. Additionally, does "non-listed in box model" mean that they were measured but not modeled? Or modeled but not measured?

**Reply:** We have added explanation of alkenes* in the caption of Table S2. Additionally, "non-listed in box model" represents that some VOCs were measured but not simulated in box model, as they are not included in the MCMv3.3.1 chemical mechanism. These have been added in the caption of Table S2 as below.

**Table S2.** Summary of the correlation coefficient of five-point calibration (i.e., 2, 4, 6, 8, 10 ppbv) for the 55 VOC species during the May and August of 2019 at the three sites in Zibo city. Alkenes* denotes anthropogenic alkenes, excluding isoprene in this study. "Non-listed in box model" represents ten measured VOC species that cannot be simulated in box model.

**Line 155-159**: "In this study, the box model (based on the Framework for 0-D Atmospheric Modeling, F0AM) (Wolfe et al., 2016) was applied and constrained by the mean diurnal profiles of meteorological data (i.e., temperature, relative humidity, and photolysis rates), 4 inorganic gases (i.e., $SO_2$, CO, NO, and $NO_2$), and 45 speciated VOCs (in MCMv3.3.1 species list; see **Table S3**)."